# Controlling worm propagation in wireless sensor networks: Through fractal-fractional mathematical perspectives

Mian Imad Shah[1], Eltigani Ismail Hassan[2], Amjad Ali[1], Abdulghani Muhyi[3,4]*, Waleed Eltayeb Ahmed[2], Khaled Aldwoah[5]*

1 Department of Mathematics and Statistics, University of Swat, Khyber Pakhtunkhwa, Pakistan, 2 Department of Mathematics and Statistics, Imam Mohammad Ibn Saud Islamic University (IMSIU), Riyadh, Saudi Arabia, 3 Department of Mathematics, Hajjah University, Hajjah, Yemen, 4 Department of Mechatronics Engineering, Faculty of Engineering and Smart Computing, Modern Specialized University, Sana'a, Yemen, 5 Department of Mathematics, Faculty of Science, Islamic University of Madinah, Madinah, Saudi Arabia

* muhyi2007@gmail.com (AM); aldwoah@iu.edu.sa (KA)

## Abstract

Wireless Sensor Networks (WSNs) are particularly vulnerable to malware attacks due to their limited processing power, memory, and energy, which makes defending against such threats especially challenging. To mitigate these serious security issues caused by malware infection, various preventive measures can be implemented, such as honeypots, robust security protocols, hardware-based protections, regular updates, firewalls, and intrusion detection systems (IDS). Considering these security concerns, we adopt an advanced version of the existing susceptible–infectious–protected–recovered **SIPR** model that incorporates a fractional-fractal derivative (FFD) defined in the Atangana-Baleanu-Caputo (ABC) sense, which offers a more realistic representation than the classical model. Furthermore, this research work introduced a new isolated nodes compartment $I_1$, along with parameters $\gamma_2$ and $\delta_1$, defining the recovery and isolation rates of $I_1$, respectively, in the existing **SIPR** model. Moreover, this study focuses on the existence and uniqueness of solutions, stability analysis, control theory and numerical approximation for the proposed generalized susceptible–infectious isolated-protected–recovered **SII$_1$PR** model. Additionally, nonlinear and fixed-point theory are used to obtain the results of existence and stability analysis. On the same line, Newton polynomial-based numerical scheme was established for the proposed modified model. The dynamics of desired results are visualized using MATLAB.

## 1 Introduction

This section of research work is committed to the fundamentals of mathematical modeling (MM), fractional calculus (FC), propagation of worms in wireless sensor networks (WSNs), literature survey, and the new findings of this research work.

**Data availability statement:** All relevant data are within the paper.

**Funding:** This work was supported and funded by the Deanship of Scientific Research at Imam Mohammad Ibn Saud Islamic University (IMSIU) (grant number IMSIU-DDRSP2502).

**Competing interests:** The authors have declared that no competing interests exist.

## 1.1 Mathematical modeling and fractional calculus

MM plays a prominent role in contemporary mathematics because it connects real-world scenarios with their mathematical representation, enabling systematic investigation and resolution of complex problems [1]. This approach allows researchers to address current challenges while predicting future possibilities [2]. MM has shown wide applications across various disciplines, ranging from engineering to biological systems. MM plays a fundamental role in studying the dynamics of worm spread by employing methodologies in epidemiological models. These models were originally developed for the study of the spread of diseases and have shown effectiveness in understanding the transmission of malware in network systems [3]. Recent studies have demonstrated the flexibility of mathematical methods to model the dynamic propagation of malware alongside the dissemination of antiviral software by considering interactions between malicious attacks and defense mechanisms in cyber environments [4]. Consequently, to address critical vulnerabilities of malicious cybersecurity attacks to networks, the analysis of viral behavior in WSNs focuses on stability and protection of networks [5]. The modeling of dynamical systems and their numerical simulations have become a foundation for advancements in engineering and scientific inquiry [6]. In particular, fractional-order operators have been extensively studied as powerful tools to extend traditional models [7]. These operators in understanding processes that exhibit hereditary properties and memory effects, which are not addressed by classical models of integer order [8].

Fractional calculus (FC) is a field of mathematics that originates and bridges theoretical and empirical investigations with real-world applications. FC has contributed a lot by addressing fundamental mathematical questions by involving appropriate numerical schemes, analytical techniques, and suitable representations across various domains [9]. Fractional derivatives (FDs) also play a crucial role in the development of modeling for frequency-dependent damping behavior in viscoelastic materials, biological phenomena, chemical processes, and engineering problems [10]. Recently, researchers have recognized some differential and integral operators prominent in capturing the complexity of inheritance, in which fractal-differential and integral operators overcome the limitations of conventional approaches [11]. The fractal-fractional derivative (FFD) has advantages over the classical derivatives, due to its important applications in the modeling of multi-faceted real-world phenomena that can not be addressed by classical ones. Contrary to classical derivatives, which focus on local behavior, FFD applies to systems with long-range dependencies or hereditary traits because it accounts for non-local effects and memory properties [12]. The merging of fractal dimensions and fractional calculus integrate the representation of irregular structures and the processes of anomalous diffusion more accurately [13]. FFD operator is helpful in disciplines such as physics, biology, and engineering because it overcomes the inadequacy of classical derivatives for processes with scaling behavior, power-law dynamics, or multi-scale interactions [14]. Also, the non-local effects of FFD improve the modeling of complex systems such as natural and engineered systems containing discontinuous or highly variable functions [15].

## 1.2 Survey literature on wireless sensor networks

The recent development in information technology has turned out all the fields of science efficiently and connectivity [16]. However, this advancement poses threats like cyber attacks, specifically to wireless sensor networks [17]. National security and safety are increasingly threatened by malicious attacks on networks [18]. Thus, ensuring the security, reliability, and efficiency of WSNs is of the highest priority. WSNs, being a key element of contemporary technological development, play a key role in achieving the desired goals, defining protocols, and gathering information [19]. Networks composed of low-cost, intelligent sensor nodes have proven highly effective in numerous applications ranging from tracking traffic flow, fuel, water conditions, seismic movements, and health parameters [20]. Similarly, these networks also play a critical role in agriculture, disaster mitigation, and environmental contaminant detection [21]. The integration of the internet with WSNs improves real-time information retrieval, facilitates quick decision-making, and improves technological efficiency [22]. However, among the several challenges that persist, one of the most significant threats is malware propagation, which compromises the confidentiality and security of data [23]. Therefore, ensuring the security of WSNs is crucial, particularly in ensuring the reliability of data, such as infrastructure management and healthcare applications. The intersection of MM, FC, and WSNs security represents a dynamic and evolving field in various research fields [24]. This convergence highlights the necessity of modern technologies for addressing the contemporary challenges related to cybersecurity and technological resilience [25].

Global models have observed noteworthy examples, including a review [26] which examines SI compartmental models based on the Kermack-McKendrick paradigm adapted for malware propagation in WSNs. This study determines that existing models have insufficiently considered energy and memory management, authentication schemes, and sensor mobility. Contemporary research on internet worm propagation relies primarily on epidemic theory [27] with state machine approaches. These classical models include: the Susceptible-Infected-Susceptible (SIS) model, the SIR model, the Two-factor model, and the Improved Worm Mitigation (IWM) model [28]. These differential equations provide a framework to effectively describe internet worm propagation characteristics to some extent. The SIR model extends the SIS model and is widely applied for analyzing internet worm dynamics. However, these models are not specially designed for WSNs and thus fail to account for energy consumption during propagation. In the SIS model [29], hosts exist in either susceptible or infectious states. A susceptible host may transit to infectious with a certain probability per unit time, while an infectious host may revert to susceptible with a certain probability per unit time (where the probability of Susceptible and infectious are not equal). At time $t$, with initial host N, the infectious host is I(t) and the susceptible host S(t). Critically, this model neglects host death from severe infections and ignores potential immunity after disinfection. The SIR model [29] addresses these limitations by adding a removed state. When an infectious host is cleaned, it transitions to removed and gains immunity; susceptible hosts become infectious with a certain probability, while infectious hosts transition to removed with a certain probability. At time t, recovered hosts R($t$) complete the state set alongside S(t) and I(t). While this accounts for immunization/isolation/death, it erroneously assumes susceptible host remain perpetually active. This observed a serious problem for WSNs as expire from energy depletion. This is due to the removed nodes remain vulnerable to new worm strains in large-scale deployments. During the investigation in [30] extends SIR for WSNs where fundamental limitations persist. Various researchers studied the propagation of worms through WSNs from different aspects and published a variety of articles. For a more detail study we referred [31–33] to the readers.

## 1.3 Generalized $SII_1PR$ model

To address the aforementioned challenges, this research work proposed a generalized $SII_1PR$ model, which overcomes the above constraints by: introducing an isolated compartment to separate infected nodes, preventing the other nodes from infection. On line by the inter-venting fractal-fractional operator instead of integer-order derivatives, which capturing memory and hereditary effects. This framework maintains mathematical tractability while accurately modeled WSNs specific dynamics like intermittent connectivity and energy-constrained propagation. The newly established proposed

model of five states, where each state represents a group of nodes: $S(t)$ which are vulnerable to malware attacks and not yet infected. These states are susceptible nodes, infected nodes $I(t)$, the nodes that are already infected and capable of spreading malware, Isolated nodes $I_1$, which are isolated from infected and protected nodes $P(t)$, which restrain the spread of malware in the network and $R(t)$ recovered nodes which spot and remove malware infections by using a detection mechanism and at any time $t$ the network comprises $N(t)$ nodes. The characteristics and properties posses by all the nodes in proposed model are the same as that of model [34]. A new compartment $I_1(t)$, representing isolated nodes, is introduced for nodes that are removed from the network to reduce the propagation worm. The isolated compartment $I_1$ reflects practical WSNs security measures where infected nodes are forcefully disconnected (but retained in the network for potential recovery).

The present study formulates the generalized $SII_1PR$ model by deriving the governing equations, qualitative analysis, control strategies, and employing numerical techniques to solve the fractal-fractional system. To evaluate the efficiency of the proposed modified model in compared to the classical model by [34], simulations studies will be conducted by considering key parameters such as infection rate, isolation effectiveness, and overall network recovery. The aims of this study are to demonstrate the enhanced resilience of WSNs against worm attacks through the proposed generalized model, offering valuable insights into robust epidemic modeling for network security. The introduction of $I_1(t)$, in the classical SIPR model, directly addresses the practical challenges related to isolating infected nodes to prevent further propagation. Finally, this study proposed a strategy not only to improve the performance of the model but also to provide meaningful insights into how to control the propagation of worms in WSNs.

### 1.4 Organization of the paper

This article is structured as follows: Sect 1 introduces fractional calculus, the framework for mathematical modeling, and the dynamics of WSN propagation. Sect 2 presents the fundamental materials, including definitions, lemmas, and theorems of FC, necessary for subsequent sections. Sect 3 details the existence and stability results for the generalized fractional WSN mathematical model. Sect 4 introduces the proposed WSN model, covering equilibrium points, the basic reproductive number, and its sensitivity analysis. Sect 5 constructs a numerical scheme based on Newton polynomial for the considered WSN model. Sect 6 provides graphical visualizations and discusses the results. Finally, the study presents the conclusion and future work.

## 2 Preliminaries

This section establishes the fundamental definitions and concepts of Fractional Calculus (FC), presenting well-known generalizations like the Gamma and Beta functions and their interconnections. We review key definitions, lemmas, and theorems relevant to our reformulated model (12) and discuss essential definitions for stability analysis. This framework supports achieving the main objectives of this investigation.

**Definition 1.** *[35] The Gamma function is defined as:*

$$\Gamma(\varsigma) = \int_0^\infty t^{\varsigma-1} e^{-t} dt, \quad where \ \ Re(\varsigma) > 0. \tag{1}$$

Certain properties related to Gamma function are mentioned in Table 1.

**Definition 2.** *[36] The Beta function is defined as:*

$$\beta(\varsigma_1, \varsigma_2) = \int_0^1 t^{\varsigma_1-1}(1-t)^{\varsigma_2-1} dt, \quad for \ \varsigma_1, \varsigma_2 \in \mathbb{C}, such \ that \ Re(\varsigma_1), Re(\varsigma_2) > 0 \tag{2}$$

**Table 1.** Properties of Gamma function.

| Property | Description |
|---|---|
| $\Gamma(\varsigma + 1) = \varsigma\Gamma(\varsigma)$ | Recurrence relation |
| $\Gamma(\varsigma) = \int_0^\infty t^{\varsigma-1} e^{-t}\, dt$ | Gamma function representation in term of an Integral |
| $\Gamma\left(\frac{1}{2}\right) = \sqrt{\pi}$ | Special value for $\Gamma\left(\frac{1}{2}\right)$ |
| $\Gamma(\varsigma) = (\varsigma - 1)!$ | Gamma function as the generalization of the factorial function |
| $\Gamma(\varsigma)\Gamma(\varsigma + 1) = \varsigma\Gamma(\varsigma)^2$ | Duplication formula |
| $\lim_{\varsigma\to\infty} \frac{\Gamma(\varsigma)}{\varsigma!} = 0$ | Asymptotic property |

Certain properties related to Beta function are mentioned in Table 2.

**Definition 3.** *[37] Consider that $\tau(\mathbf{t})$ is a continuous function, then the fractal derivative of order $\varsigma$ of $\tau(\mathbf{t})$ is given as:*

$$\frac{d\tau(\mathbf{t})}{dt^\varsigma} = \lim_{t_1{}^\varsigma \to t^\varsigma} \frac{\tau(\boldsymbol{t_1}) - \tau(\mathbf{t})}{t_1{}^\varsigma - t^\varsigma}, \quad \text{for} \quad \varsigma > 0, \tag{3}$$

*where its fractional generalized form is given by*

$$\frac{d^{\varsigma_1}\tau(\mathbf{t})}{dt^{\varsigma_2}} = \lim_{t_1{}^{\varsigma_2} \to t^{\varsigma_2}} \frac{\tau^{\varsigma_1}(t_1) - \tau^{\varsigma_1}(t)}{t_1{}^{\varsigma_2} - t^{\varsigma_2}}, \quad \text{for} \quad \varsigma_1, \varsigma_2 > 0. \tag{4}$$

**Remark 1.** *Let $\tau(\mathbf{t}) = t$, where both of its fractional-order and fractal fractional order derivatives exist, we have*

$$\frac{dt}{dt^\varsigma} = \frac{d\tau}{dt} \cdot \frac{dt}{dt^\varsigma} = \frac{1}{\varsigma} t^{1-\varsigma} \frac{d\tau}{dt}, \quad \text{by using the substitution} \quad dt^\varsigma = \varsigma t^{\varsigma-1} dt. \tag{5}$$

**Definition 4.** *[38] Let $\tau(\mathbf{t})$ be continuous on $I = [0, T]$, with $\varsigma_2 \geq 0$ and $\varsigma_1 \in (0, 1]$, then the fractal fractional ABC derivative ($^{\text{F−ABC}}\mathbb{D}$) of $\tau(\mathbf{t})$ is defined as:*

$$^{\text{F−ABC}}\mathbb{D}_{0,t}^{\varsigma_1,\varsigma_2}\tau(\mathbf{t}) = \frac{\mathbb{ABC}(\varsigma_1)}{1 - \varsigma_1} \frac{d}{dt^{\varsigma_2}} \int_0^t \mathbb{E}_{\varsigma_1}\left(-\frac{\varsigma_1}{1 - \varsigma_1}(t - s)^{\varsigma_1}\right)\tau(\boldsymbol{s})ds. \tag{6}$$

**Table 2.** Properties of Beta function.

| Property | Description |
|---|---|
| $\beta(\varsigma_1, \varsigma_2) = \beta(\varsigma_2, \varsigma_1)$ | Symmetry property |
| $\beta(\varsigma_1, \varsigma_2) = \frac{\Gamma(\varsigma_1)\Gamma(\varsigma_2)}{\Gamma(\varsigma_1+\varsigma_2)}$ | Relation of Gamma and Beta functions |
| $\beta(\varsigma_1, \varsigma_2) = \frac{1}{\varsigma_2-1}\int_0^1 t^{\varsigma_1-1}(1 - t)^{\varsigma_2-2}\, dt$ | Formula of reduction |
| $\beta(\varsigma_1, \varsigma_2) = 2\int_0^{\pi/2}(\sin t)^{2\varsigma_1-1}(\cos t)^{2\varsigma_2-1}\, dt$ | Trigonometric representation |
| $\beta(\varsigma_1, \varsigma_2) = \frac{\Gamma(\varsigma_1)\Gamma(\varsigma_2)}{\Gamma(\varsigma_1+\varsigma_2)} = \int_0^\infty \frac{t^{\varsigma_1-1}}{(1+t)^{\varsigma_1+\varsigma_2}}\, dt$ | Representation of Beta function in terms of integral |

**Definition 5.** [38] Let the function $\tau(t)$ be continuous on $I = [0, T]$, such that $\varsigma_2 \geq 0$ and $\varsigma_1 \in (0, 1]$, then the fractal-fractional integral (FFI) in the sense of $F - ABC$ is follows as:

$$^{F-ABC}\mathbb{I}_{0,t}^{\varsigma_1,\varsigma_2}\tau(t) = \frac{\varsigma_1\varsigma_2}{\mathbb{ABC}(\varsigma_1)\Gamma(\varsigma_1)} \int_0^t (t-s)^{\varsigma_1}s^{\varsigma_2-1}\tau(s)ds + \frac{\varsigma_2(1-\varsigma_1)t^{\varsigma_2-1}\tau(t)}{\mathbb{ABC}(\varsigma_1)}, \tag{7}$$

where $\varsigma_1 > 0$, $\varsigma_2 \leq 1$, and $\varsigma_2 \in \mathbb{N}$. The function $\mathbb{ABC}(\varsigma_1)$ is defined as:

$$\mathbb{ABC}(\varsigma_1) = 1 - \varsigma_1 + \frac{\varsigma_1}{\Gamma(\varsigma_1)}. \tag{8}$$

**Lemma 1.** For any function $\mathfrak{I}(t) \in C(I) \cap C(I)$, the solution of

$$^{\mathbb{FF}-\mathbb{ABC}}\mathbb{D}^{\varsigma_1,\varsigma_2}\mathfrak{I}(t) = \mathfrak{I}(t), \quad \text{for} \quad 0 < \varsigma_1, \ \varsigma_2 < 1,$$
$$\mathfrak{I}(0) = \mathfrak{I}_0, \tag{9}$$

is given by

$$\mathfrak{I}(t) = \mathfrak{I}_0 + \frac{\varsigma_1\varsigma_2}{\mathbb{ABC}(\varsigma_1)\Gamma(\varsigma_1)} \int_0^t (t-s)^{\varsigma_1}s^{\varsigma_2-1}\mathfrak{I}(s)ds + \frac{\varsigma_2(1-\varsigma_1)t^{\varsigma_2-1}\mathfrak{I}(t)}{\mathbb{ABC}(\varsigma_1)}. \tag{10}$$

**Definition 6.** A function $\mathbb{F}$ with a Lipschitz constant r is Lipschitz continuous (LC) on $\mathbb{R}$, if

$$\|\mathbb{F}_1(Y_1(t)) - \mathbb{F}_1(Y_2(t))\| \leq r\|Y_1 - Y_2\|, \quad \forall \ Y_1, Y_2 \in \mathbb{R}. \tag{11}$$

**Theorem 1.** Let $\aleph$ be a convex closed subset of a Banach space $\mathbb{X}$, then $\mathbb{F} : \mathbb{X} \to \mathbb{X}$ is completely continuous and has at least one fixed point.

**Assumption 1.** Consider the functions $\tau_1(t)$ and $\tau_2(t)$. The following norm properties hold:

1. $\|\tau_1(t) + \tau_2(t)\| \leq \|\tau_1(t)\| + \|\tau_2(t)\|$,
2. $\|\tau_1(t) \cdot \tau_2(t)\| = \|\tau_2(t)\| \cdot \|\tau_1(t)\|$,
3. $\|\tau_1(t) + \tau_2(t)\| = \|\tau_2(t) + \tau_1(t)\|$,
4. $\|\tau_1(t) - \tau_2(t)\| = \|\tau_2(t) - \tau_1(t)\|$.

## 3 Investigation of generalized WSNs model

In this section, we investigate the proposed generalized model (12) regarding existence and stability. A comprehensive literature review highlights the significance of worm propagation in WSNs and reveals that this research area requires further attention to explore its various aspects. Our study integrates a FFD in the ABC sense into the existing SIPR framework, achieving enhanced realism compared to classical models. Furthermore, we extend the model [34] by introducing a new compartment $I_1$ for isolated nodes, accompanied by parameters $\gamma_2$ (recovery rate) and $\delta_1$ (isolation rate) for $I_1$. The advanced $SII_1PR$ model is formulated as follows:

$$^{F\text{-}ABC}\mathbb{D}_{0,t}^{\varsigma_1,\varsigma_2}S(t) = (1-\rho)\mu - \beta\frac{I(t)S(t)}{N(t)} - \mu S(t),$$

$$^{F\text{-}ABC}\mathbb{D}_{0,t}^{\varsigma_1,\varsigma_2}I(t) = \beta\frac{I(t)S(t)}{N(t)} - (\gamma_1 + \mu + \gamma + \delta_1)I(t),$$

$$^{F\text{-}ABC}\mathbb{D}_{0,t}^{\varsigma_1,\varsigma_2}I_1(t) = \delta_1 I(t) - (\mu + \gamma_2)I_1(t),$$

$$\text{F-ABC}\mathbb{D}_{0,t}^{\varsigma_1,\varsigma_2}P(t) = \mu\rho - \mu P(t),$$

$$\text{F-ABC}\mathbb{D}_{0,t}^{\varsigma_1,\varsigma_2}R(t) = \gamma_1 I(t) - \mu R(t) + \gamma_2 I_1(t), \tag{12}$$

with $S_0 = S(0)$, $I_0 = I(0)$, $P_0 = P(0)$, $I_1(0) = I_{10}$, $R_0 = R(0)$ with $S(t) + I(t) + P(t) + I_1(t) + R(t) = 1 = N(t)$.

Fig 1 represents the flow chart to describe the dynamics of our proposed model 12) is follow as:

The descriptions of the parameters used in the generalized $SII_1PR$ model is given in Table 3.

## 3.1 Results of existence

We define the following system of operator equations for the proposed model (12) as follows:

$$\mathbb{K}_1(S, t) = (1 - \rho)\mu - \beta\frac{S(t)I(t)}{N(t)} - \mu S(t),$$

$$\mathbb{K}_2(I, t) = \beta\frac{S(t)I(t)}{N(t)} - (\gamma_1 + \mu + \gamma + \delta_1)I(t),$$

$$\mathbb{K}_3(I_1, t) = \delta_1 I(t) - (\mu + \gamma_2)I_1(t),$$

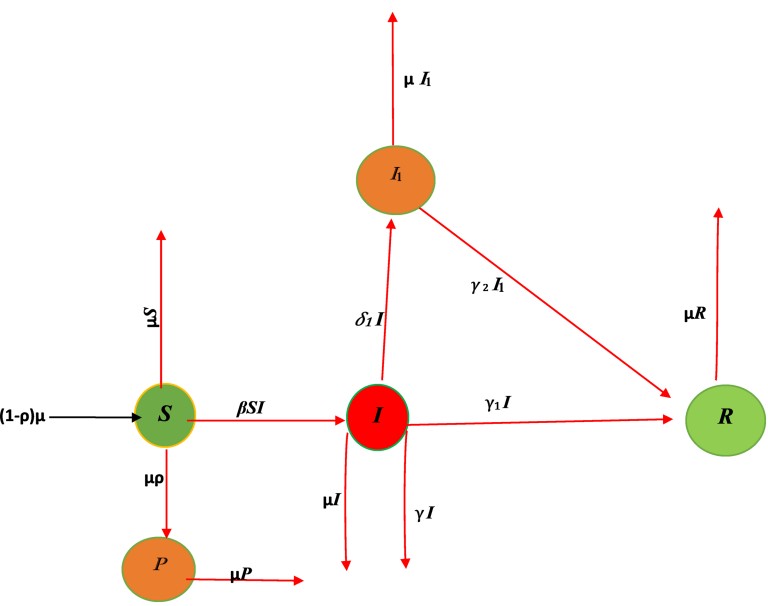

**Fig 1**. Flow chart for model (12).

**Table 3**. Description of parameters involved in Model (12).

| Parameters | Representation of parameters |
|---|---|
| $\rho$ | protection rate of $P(t)$ |
| $\beta$ | contact rate due to which infection causes |
| $\mu$ | natural death rate |
| $\gamma_1$ | recovery rate of $I(t)$ |
| $\gamma$ | death rate of infected nodes |
| $\gamma_2$ | recovery rate of $I_1(t)$ |
| $\delta_1$ | rate of isolation. |

$$\mathbb{K}_4(P, t) = \mu\rho - \mu P(t),$$
$$\mathbb{K}_5(R, t) = \gamma_1 I(t) - \mu R(t) + \gamma_2 I_1(t). \tag{13}$$

Using the concepts of FF-order derivatives from Definition 5 with the combination of system of operator Eq (13) in proposed model (12), we get

$$^{\mathrm{ABC}}\mathbb{D}_{0,t}^{\varsigma_1} S(t) = \varsigma_2 t^{\varsigma_2-1} \mathbb{K}_1(S, t),$$
$$^{\mathrm{ABC}}\mathbb{D}_{0,t}^{\varsigma_1} I(t) = \varsigma_2 t^{\varsigma_2-1} \mathbb{K}_2(I, t),$$
$$^{\mathrm{ABC}}\mathbb{D}_{0,t}^{\varsigma_1} I_1(t) = \varsigma_2 t^{\varsigma_2-1} \mathbb{K}_3(I_1(t), t),$$
$$^{\mathrm{ABC}}\mathbb{D}_{0,t}^{\varsigma_1} P(t) = \varsigma_2 t^{\varsigma_2-1} \mathbb{K}_4(P, t),$$
$$^{\mathrm{ABC}}\mathbb{D}_{0,t}^{\varsigma_1} R(t) = \varsigma_2 t^{\varsigma_2-1} \mathbb{K}_5(R, t). \tag{14}$$

**Theorem 2.** *The solutions of the proposed generalized model (12) are non-negative and bounded.*

*Proof*: From the considered model (12), we have

$$^{\mathrm{F-ABC}}\mathbb{D}_{0,t}^{\varsigma_1,\varsigma_2} S(t)\Big|_{S=0} = \mu(-\rho + 1) \geq 0,$$
$$^{\mathrm{F-ABC}}\mathbb{D}_{0,t}^{\varsigma_1,\varsigma_2} I(t)\Big|_{I=0} = 0,$$
$$^{\mathrm{F-ABC}}\mathbb{D}_{0,t}^{\varsigma_1,\varsigma_2} I_1(t)(t)\Big|_{I_1(t)=0} = I(t)\delta_1 \quad \geq \quad 0,$$
$$^{\mathrm{F-ABC}}\mathbb{D}_{0,t}^{\varsigma_1,\varsigma_2} P(t)\Big|_{P=0} = \quad \rho\mu \quad \geq 0,$$
$$^{\mathrm{F-ABC}}\mathbb{D}_{0,t}^{\varsigma_1,\varsigma_2} R(t)\Big|_{R=0} = I_1(t)(t)\gamma_2 + \gamma_1 I(t) \geq 0. \tag{15}$$

Thus, for $\left(S(0), I(0), I_1(0), P(0), R(0)\right) \in \mathbb{R}_+^5$, $R^5$ is a positive invariant set and therefore every hyperplane is non-negative and bounded. ☐

**Theorem 3.** *The operators defined in system of Eq (13), namely $\mathbb{K}_1(S, t)$, $\mathbb{K}_2(I, t)$, $\mathbb{K}_4(P, t)$, $\mathbb{K}_3(I_1, t)$ and $\mathbb{K}_5(R, t)$ satisfy the Lipschitz condition (LC) and exhibit contraction properties.*

*Proof*: Consider two solutions $S_1$ and $S_2$ of first compartment of considered model (12) for kernel $\mathbb{K}_1$, we have

$$\mathbb{K}_1(S_1, t) = (-\rho + 1)\mu - \beta \frac{I(t)S_1(t)}{N(t)} - S_1(t)\mu,$$
$$\mathbb{K}_1(S_2, t) = \mu(1 - \rho) - \beta \frac{I(t)S_2(t)}{N(t)} - S_2(t)\mu.$$

Subtracting $\mathbb{K}_2$ and $\mathbb{K}_1$ and applying the maximum norm, we obtain

$$\|\mathbb{K}_1(S_1, t) - \mathbb{K}_1(S_2, t)\| = \left\| \mu(-\rho + 1) - \beta \frac{I(t)S_1(t)}{N(t)} - S_1(t)\mu \right.$$
$$\left. - \{(-\rho + 1)\mu - \beta \frac{I(t)S_2(t)}{N(t)} - S_2(t)\mu\} \right\|$$
$$= \left\| -\beta \frac{I(t)S_1(t)}{N(t)} - \mu S_1(t) + \beta \frac{I(t)S_2(t)}{N(t)} + \mu S_2(t) \right\|$$

$$= \left\| \left( \beta \frac{I(t)}{N(t)} + \mu \right)(S_2(t) - S_1(t)) \right\|$$

$$= \left\| \beta \frac{I(t)}{N(t)} + \mu \right\| \left\| S_2(t) - S_1(t) \right\|$$

$$\leq \mu + \left\| \beta \frac{I(t)}{N(t)} \right\| \left\| S_2(t) - S_1(t) \right\|$$

$$\leq \Psi_1 \| S_2(t) - S_1(t) \|.$$

Defining $\Psi_1 = \mu + \left\| \frac{\beta I(t)}{N(t)} \right\| < 1$, it follows that the operator $\mathbb{K}_1$ satisfies the Lipschitz condition (LC). Similarly, we can show that the remaining operator also satisfies the LC. □

**Theorem 4.** *The solution of proposed $SII_1PR$ model ([12]) is unique, if the following inequality holds true*

$$\left( \frac{\varsigma_2(1-\varsigma_1)}{\mathbb{ABC}(\varsigma_1)} + \frac{\varsigma_1\varsigma_2\Gamma(\varsigma_2)}{\mathbb{ABC}(\varsigma_1)\Gamma(\varsigma_1+\varsigma_2)} \mathfrak{I}_i \right) \leq 1, \qquad i \in \mathbb{N}_i^5. \tag{16}$$

*Proof*: Let us consider the first compartment of our considered model ([12])

$${}^{\text{F-ABC}}\mathbb{D}_{0,t}^{\varsigma_1,\varsigma_2} S(t) = \mu(-\rho + 1) - \beta \frac{I(t)S(t)}{N(t)} - S(t)\mu. \tag{17}$$

Applying the operator ${}^{\text{F-ABC}}\mathbb{I}_{0,t}^{\varsigma_1,\varsigma_2}$ to both sides of [Eq (17)], we obtain

$$S(t) = S(0) + {}^{\text{F-ABC}}\mathbb{I}_{0,t}^{\varsigma_1,\varsigma_2} \left( \mathbb{K}_1\left(S(t)\right) \right).$$

Similarly, we can obtain the remaining compartments of model ([12]).

$$I(t) = I(0) + {}^{\text{F-ABC}}\mathbb{I}_{0,t}^{\varsigma_1,\varsigma_2} \left( \mathbb{K}_2\left(I(t)\right) \right),$$

$$I_1(t) = I_1(0) + {}^{\text{F-ABC}}\mathbb{I}_{0,t}^{\varsigma_1,\varsigma_2} \left( \mathbb{K}_3\left(I_1(t)\right) \right),$$

$$P(t) = P(0) + {}^{\text{F-ABC}}\mathbb{I}_{0,t}^{\varsigma_1,\varsigma_2} \left( \mathbb{K}_4\left(P(t)\right) \right),$$

$$R(t) = R(0) + {}^{\text{F-ABC}}\mathbb{I}_{0,t}^{\varsigma_1,\varsigma_2} \left( \mathbb{K}_4(R(t)) \right).$$

Consider that $\mathbb{F}_1, \mathbb{F}_2, \mathbb{F}_3, \mathbb{F}_4$, and $\mathbb{F}_5$ are Picard operators, we have

$$\mathbb{F}_1 S(t) = S(0) + {}^{\text{F-ABC}}\mathbb{I}_{0,t}^{\varsigma_1,\varsigma_2} \left\{ \mathbb{K}_1\left(S(t)\right) \right\},$$

$$\mathbb{F}_2 I(t) = I(0) + {}^{\text{F-ABC}}\mathbb{I}_{0,t}^{\varsigma_1,\varsigma_2} \left\{ \mathbb{K}_2\left(I(t)\right) \right\},$$

$$\mathbb{F}_3 I_1(t) = I_1(0) + {}^{\text{F-ABC}}\mathbb{I}_{0,t}^{\varsigma_1,\varsigma_2} \left\{ \mathbb{K}_3\left(I_1(t)\right) \right\},$$

$$\mathbb{F}_4 P(t) = P(0) + {}^{\text{F-ABC}}\mathbb{I}_{0,t}^{\varsigma_1,\varsigma_2} \left\{ \mathbb{K}_4\left(P(t)\right) \right\},$$

$$\mathbb{F}_5 R(t) = R(0) + {}^{\text{F-ABC}}\mathbb{I}_{0,t}^{\varsigma_1,\varsigma_2} \left\{ \mathbb{K}_5\left(R(t)\right) \right\}. \tag{18}$$

Here, we present boundedness results for the proposed Picard operators as follows:

$$\left\|\mathbb{F}_1 S(t)\right\| = \left\|S(0) + {}^{F-ABC}\mathbb{I}_{0,t}^{\varsigma_1,\varsigma_2}\left\{\mathbb{K}_1\left(S(t)\right)\right\}\right\|$$
$$\leq S(0) + {}^{F-ABC}\mathbb{I}_{0,t}^{\varsigma_1,\varsigma_2}\left\|\mathbb{K}_1\left(S(t)\right)\right\|,$$

$$\left\|\mathbb{F}_2 I(t)\right\| = \left\|I(0) + {}^{F-ABC}\mathbb{I}_{0,t}^{\varsigma_1,\varsigma_2}\left\{\mathbb{K}_2\left(I(t)\right)\right\}\right\|$$
$$\leq I(0) + {}^{F-ABC}\mathbb{I}_{0,t}^{\varsigma_1,\varsigma_2}\left\|\mathbb{K}_2\left(I(t)\right)\right\|,$$

$$\left\|\mathbb{F}_3 I_1(t)\right\| = \left\|I_1(0) + {}^{F-ABC}\mathbb{I}_{0,t}^{\varsigma_1,\varsigma_2}\left\{\mathbb{K}_3(I_1(t))\right\}\right\|$$
$$\leq I_1(0) + {}^{F-ABC}\mathbb{I}_{0,t}^{\varsigma_1,\varsigma_2}\left\|\mathbb{K}_3(I_1(t))\right\|,$$

$$\left\|\mathbb{F}_4 P(t)\right\| = \left\|P(0) + {}^{F-ABC}\mathbb{I}_{0,t}^{\varsigma_1,\varsigma_2}\left\{\mathbb{K}_4\left(P(t)\right)\right\}\right\|$$
$$\leq P(0) + {}^{F-ABC}\mathbb{I}_{0,t}^{\varsigma_1,\varsigma_2}\left\|\mathbb{K}_4\left(P(t)\right)\right\|,$$

$$\left\|\mathbb{F}_5 R(t)\right\| = \left\|R(0) + {}^{F-ABC}\mathbb{I}_{0,t}^{\varsigma_1,\varsigma_2}\left\{\mathbb{K}_5\left(R(t)\right)\right\}\right\|$$
$$\leq R(0) + {}^{F-ABC}\mathbb{I}_{0,t}^{\varsigma_1,\varsigma_2}\left\|\mathbb{K}_5\left(R(t)\right)\right\|. \qquad (19)$$

Since $\mathbb{K}(S, I, I_1, P, R, t)$ is LC, it implies that $\mathbb{K}(S, I, I_1, P, R, t)$ is bounded.
Thus, there exist $\ell_1, \ell_2, \ell_3, \ell_4,$ and $\ell_5$ constants such that

$$\|\mathbb{K}_1(S, t)\| \leq \ell_1, \quad \|\mathbb{K}_2(I, t)\| \leq \ell_2, \quad \|\mathbb{K}_3(I_1, t)\| \leq \ell_3,$$
$$\|\mathbb{K}_4(P, t)\| \leq \ell_4, \quad \|\mathbb{K}_5(R, t)\| \leq \ell_5.$$

Thus,

$$\left\|\mathbb{F}_1 S(t)\right\| \leq S(0) + {}^{F-ABC}\mathbb{I}_{0,t}^{\varsigma_1,\varsigma_2}\left\|\mathbb{K}_1\left(S(t)\right)\right\|,$$
$$\leq S(0) + {}^{F-ABC}\mathbb{I}_{0,t}^{\varsigma_1,\varsigma_2}\left(\ell_1\right),$$
$$= S(0) + \left(\ell_1 {}^{F-ABC}\mathbb{I}_{0,t}^{\varsigma_1,\varsigma_2}(1)\right),$$
$$\leq S(0) + \ell_1\left[\frac{\varsigma_1\varsigma_2}{\mathbb{ABC}(\varsigma_1)\Gamma(\varsigma_1)}\int_0^t (t-\tau)^{\varsigma_1-1}(1)d\tau + \frac{\varsigma_2(1-\varsigma_1)t^{\varsigma_2-1}}{\mathbb{ABC}(\varsigma_1)}(1)\right]. \qquad (20)$$

For simplicity, putting the maximum value of $t$ to obtain beta function.
Now, assume that $\exists \quad \mathbb{Y}_1 \in R$, such that $\mathbb{Y}_1 \geq t \geq 0$:

$$\left\|\mathbb{F}_1 S(t)\right\| \leq S(0) + \ell_1\left[\frac{\varsigma_1\varsigma_2}{\mathbb{ABC}(\varsigma_1)\Gamma(\varsigma_1)}\int_0^1 (1-\tau)^{\varsigma_1-1}(1)d\tau + \frac{\varsigma_2(1-\varsigma_1)\mathbb{Y}_1^{\varsigma_2-1}}{\mathbb{ABC}(\varsigma_1)}\right],$$
$$= S(0) + \ell_1\left[\frac{\varsigma_1\varsigma_2}{\mathbb{ABC}(\varsigma_1)\Gamma(\varsigma_1)} \times \mathbb{B}(\varsigma_2, \varsigma_1) + \frac{\varsigma_2(1-\varsigma_1)\mathbb{Y}_1^{\varsigma_2-1}}{\mathbb{ABC}(\varsigma_1)}\right],$$

$$= S(0) + \ell_1 \left[ \frac{\varsigma_1 \varsigma_2}{\mathbb{ABC}(\varsigma_1)\Gamma(\varsigma_1)} \times \frac{\Gamma(\varsigma_2)\Gamma(\varsigma_1)}{\Gamma(\varsigma_1 + \varsigma_2)} + \frac{\varsigma_2(1 - \varsigma_1)\mathbb{Y}_1^{\varsigma_2-1}}{\mathbb{ABC}(\varsigma_1)} \right],$$

implies that

$$\left\| \mathbb{F}_1 S(t) \right\| \leq S(0) + \ell_1 \left[ \frac{\varsigma_1 \varsigma_2}{\mathbb{ABC}(\varsigma_1)} \frac{\Gamma(\varsigma_2)}{\Gamma(\varsigma_1 + \varsigma_2)} + \frac{\varsigma_2(1 - \varsigma_1)\mathbb{Y}_1^{\varsigma_2-1}}{\mathbb{ABC}(\varsigma_1)} \right].$$

Similarly, for $\mathbb{Y}_2, \mathbb{Y}_3, \mathbb{Y}_4, \mathbb{Y}_5 \in \mathbb{R}$, where $0 \leq t \leq \mathbb{Y}_2, \mathbb{Y}_3, \mathbb{Y}_4, \mathbb{Y}_5$, we have

$$\left\| \mathbb{F}_2 I(t) \right\| \leq I(0) + \ell_1 \left[ \frac{\varsigma_1 \varsigma_2}{\mathbb{ABC}(\varsigma_1)} \frac{\Gamma(\varsigma_2)}{\Gamma(\varsigma_1 + \varsigma_2)} + \frac{\varsigma_2(1 - \varsigma_1)\mathbb{Y}_2^{\varsigma_2-1}}{\mathbb{ABC}(\varsigma_1)} \right],$$

$$\left\| \mathbb{F}_3 I_1(t) \right\| \leq I_1(0) + \ell_1 \left[ \frac{\varsigma_1 \varsigma_2}{\mathbb{ABC}(\varsigma_1)} \frac{\Gamma(\varsigma_2)}{\Gamma(\varsigma_1 + \varsigma_2)} + \frac{\varsigma_2(1 - \varsigma_1)\mathbb{Y}_3^{\varsigma_2-1}}{\mathbb{ABC}(\varsigma_1)} \right],$$

$$\left\| \mathbb{F}_4 P(t) \right\| \leq P(0) + \ell_1 \left[ \frac{\varsigma_1 \varsigma_2}{\mathbb{ABC}(\varsigma_1)} \frac{\Gamma(\varsigma_2)}{\Gamma(\varsigma_1 + \varsigma_2)} + \frac{\varsigma_2(1 - \varsigma_1)\mathbb{Y}_4^{\varsigma_2-1}}{\mathbb{ABC}(\varsigma_1)} \right],$$

$$\left\| \mathbb{F}_5 R(t) \right\| \leq R(0) + \ell_1 \left[ \frac{\varsigma_1 \varsigma_2}{\mathbb{ABC}(\varsigma_1)} \frac{\Gamma(\varsigma_2)}{\Gamma(\varsigma_1 + \varsigma_2)} + \frac{\varsigma_2(1 - \varsigma_1)\mathbb{Y}_5^{\varsigma_2-1}}{\mathbb{ABC}(\varsigma_1)} \right].$$

Therefore, the operators $\mathbb{F}_1, \mathbb{F}_2, \mathbb{F}_3, \mathbb{F}_4$, and $\mathbb{F}_5$ are Picard bounded operators.

The operators $\mathbb{F}_1, \mathbb{F}_2, \mathbb{F}_3, \mathbb{F}_4$, and $\mathbb{F}_5$ satisfy the contraction principle (CP), if

$$\|\mathbb{F}_1(Y_1(t)) - \mathbb{F}_1(Y_2(t))\| \leq r\|Y_1 - Y_2\|. \tag{21}$$

Let $0 < r < 1$.

$$\|\mathbb{F}_1(S_1(t)) - \mathbb{F}_1(S_2(t))\| = \|{}^{F-ABC}\mathbb{I}_{0,t}^{\varsigma_1,\varsigma_2}\mathbb{K}_1(S_1(t)) - {}^{F-ABC}\mathbb{I}_{0,t}^{\varsigma_1,\varsigma_2}\mathbb{K}_1(S_2(t))\|,$$

$$\leq \|\mathbb{K}_1(S_1(t)) - \mathbb{K}_1(S_2(t))\|{}^{F-ABC}\mathbb{I}_{0,t}^{\varsigma_1,\varsigma_2}(1).$$

Now, putting the values of $\mathbb{K}_1(S_1, t)$ and $\mathbb{K}_1(S_2, t)$, we get

$$\|\mathbb{F}_1(S_1(t)) - \mathbb{F}_1(S_2(t))\| \leq \|\mu(-\rho + 1) - \beta \frac{I(t)S_1(t)}{N(t)} - S_1(t)\mu$$

$$- [(-\rho + 1)\mu - \beta \frac{I(t)S_2(t)}{N(t)} - S_2(t)\mu]\|{}^{F-ABC}\mathbb{I}_{0,t}^{\varsigma_1,\varsigma_2}(1),$$

$$\leq \left( \left\| \beta \frac{I(t)}{N(t)} \right\| + \mu \right) \left\| S_2(t) - S_1(t) \right\|{}^{F-ABC}\mathbb{I}_{0,t}^{\varsigma_1,\varsigma_2}(1).$$

Let $\|\beta \frac{I(t)}{N(t)}\|$ is bounded by $\varphi$, i.e., $\|\beta \frac{I(t)}{N(t)}\| \leq \varphi$. From the definition of $\mathbb{FF} - \mathbb{ABC}(1)$ for any constant, we arrive at

$$\|\mathbb{F}_1(S_1(t)) - \mathbb{F}_1(S_2(t))\| \leq \left( \mu + \left\| \beta \frac{I(t)}{N(t)} \right\| \right) \cdot \|S_2(t) - S_1(t)\|{}^{F-ABC}\mathbb{I}_{0,t}^{\varsigma_1,\varsigma_2}(1),$$

$$\leq (\mu + \varphi) \|S_1(t) - S_2(t)\| \left[ \frac{\varsigma_1 \varsigma_2}{\mathbb{ABC}(\varsigma_1)} \frac{\Gamma(\varsigma_2)}{\Gamma(\varsigma_1 + \varsigma_2)} + \frac{\varsigma_2(1 - \varsigma_1)\mathbb{Y}_1^{\varsigma_2-1}}{\mathbb{ABC}(\varsigma_1)} \right].$$

Let

$$K_1 = \left[ \frac{\varsigma_1 \varsigma_2}{\mathbb{ABC}(\varsigma_1)} \frac{\Gamma(\varsigma_2)}{\Gamma(\varsigma_1 + \varsigma_2)} + \frac{\varsigma_2(1 - \varsigma_1)\mathbb{V}_1^{\varsigma_2 - 1}}{\mathbb{ABC}(\varsigma_1)} \right] (\mu + \varphi).$$

Then

$$\|\mathbb{F}_1(S_1(t)) - \mathbb{F}_1(S_2(t))\| \leq K_1 \|S_1(t) - S_2(t)\|.$$

If $0 < K_1 < \frac{1}{\mu + \varphi}$, then $\mathbb{F}_1$ satisfies the CP.
Following the same procedure, we have
For $0 < K_2 < \frac{1}{\gamma_1 + \gamma + \mu + \delta_1 + \varphi_1}$, where

$$K_2 = \left[ \frac{\varsigma_1 \varsigma_2}{\mathbb{ABC}(\varsigma_1)} \frac{\Gamma(\varsigma_2)}{\Gamma(\varsigma_1 + \varsigma_2)} + \frac{\varsigma_2(1 - \varsigma_1)\mathbb{V}_2^{\varsigma_2 - 1}}{\mathbb{ABC}(\varsigma_1)} \right] (\gamma_1 + \gamma + \mu + \delta_1 + \varphi_1),$$

the operator $\mathbb{F}_2$ satisfies the CP.
Similarly, for $0 < K_3 < \frac{1}{\gamma_2 + \mu}$, where

$$K_3 = (\gamma_2 + \mu) \left[ \frac{\varsigma_1 \varsigma_2}{\mathbb{ABC}(\varsigma_1)} \frac{\Gamma(\varsigma_2)}{\Gamma(\varsigma_1 + \varsigma_2)} + \frac{\varsigma_2(1 - \varsigma_1)\mathbb{V}_3^{\varsigma_2 - 1}}{\mathbb{ABC}(\varsigma_1)} \right],$$

the operator $\mathbb{F}_3$ satisfies the CP.
For

$$K_4 = \mu \left[ \frac{\varsigma_1 \varsigma_2}{\mathbb{ABC}(\varsigma_1)} \frac{\Gamma(\varsigma_2)}{\Gamma(\varsigma_1 + \varsigma_2)} + \frac{\varsigma_2(1 - \varsigma_1)\mathbb{V}_4^{\varsigma_2 - 1}}{\mathbb{ABC}(\varsigma_1)} \right],$$

and

$$K_5 = \mu \left[ \frac{\varsigma_1 \varsigma_2}{\mathbb{ABC}(\varsigma_1)} \frac{\Gamma(\varsigma_2)}{\Gamma(\varsigma_1 + \varsigma_2)} + \frac{\varsigma_2(1 - \varsigma_1)\mathbb{V}_5^{\varsigma_2 - 1}}{\mathbb{ABC}(\varsigma_1)} \right],$$

where $0 < K_4, K_5 < \frac{1}{\mu}$, the operators $\mathbb{F}_4$ and $\mathbb{F}_5$ satisfy the CP.

Now, to show that the solution of (12) is unique, assuming any two solutions $S_1(t)$ and $S_2(t)$ for the first compartment of model (12), we have

$$S_1 - S_0 = {}^{F-ABC}\mathbb{I}_{0,t}^{\varsigma_1,\varsigma_2} \mathbb{K}_1(S_1,(t)),$$
$$S_2 - S_0 = {}^{F-ABC}\mathbb{I}_{0,t}^{\varsigma_1,\varsigma_2} \mathbb{K}_1(S_2,(t)).$$

Thus

$$\|S_1 - S_0 - (S_2 - S_0)\| = \left\| {}^{F-ABC}\mathbb{I}_{0,t}^{\varsigma_1,\varsigma_2} \mathbb{K}_1(S_1,(t)) - {}^{F-ABC}\mathbb{I}_{0,t}^{\varsigma_1,\varsigma_2} \mathbb{K}_1(S_2,(t)) \right\|,$$

$$\|S_1 - S_2\| \leq \left\|\mathbb{K}_1(S_1, t) - \mathbb{K}_2(S_2, (t))\right\|^{F-ABC} \mathbb{I}_{0,t}^{\varsigma_1,\varsigma_2}(1). \tag{22}$$

From system of Eq (13), we have

$$\mathbb{K}_1(S_1, t) = (-\rho + 1)\mu - \beta \frac{I(t)S_1(t)}{N(t)} - S_1(t)\mu,$$

$$\mathbb{K}_1(S_2, t) = (-\rho + 1)\mu - \beta \frac{I(t)S_2(t)}{N(t)} - S_2(t)\mu.$$

Putting $\mathbb{K}_1(S_1, t)$ and $\mathbb{K}_1(S_2, t)$ into (22), we obtain

$$\left\|S_1 - S_2\right\| \leq \left\|\mathbb{K}_1(S_1, (t)) - \mathbb{K}_2(S_2, (t))\right\|^{F-ABC} \mathbb{I}_{0,t}^{\varsigma_1,\varsigma_2}(1)$$

$$= \left\|(-\rho + 1)\mu - \beta \frac{I(t)S_1(t)}{N(t)} - S_1(t)\mu \right.$$

$$\left. - \left[(-\rho + 1)\mu - \beta \frac{I(t)S_2(t)}{N(t)} - S_2(t)\mu\right]\right\|^{F-ABC} \mathbb{I}_{0,t}^{\varsigma_1,\varsigma_2}(1)$$

$$= \left\|\left(\mu + \beta \frac{I(t)}{N(t)}\right)(S_2(t) - S_1(t))\right\|^{F-ABC} \mathbb{I}_{0,t}^{\varsigma_1,\varsigma_2}(1)$$

$$\leq \left\|\beta \frac{I(t)}{N(t)} + \mu\right\| . \left\|S_2(t) - S_1(t)\right\|^{F-ABC} \mathbb{I}_{0,t}^{\varsigma_1,\varsigma_2}(1)$$

$$= \left\|\mu\right\| + \left\|\beta \frac{I(t)}{N(t)}\right\| . \left\|S_2(t) - S_1(t)\right\|^{F-ABC} \mathbb{I}_{0,t}^{\varsigma_1,\varsigma_2}(1). \tag{23}$$

Since $\mu$ is a constant and $\left\|\beta \frac{I(t)}{N(t)}\right\|$ is bounded by $\eta_1$, we have $\left\|\beta \frac{I(t)}{N(t)}\right\| < \eta_1$ and $\|\mu\| = \mu$. Substituting these values into (23), we obtain

$$\left\|S_1(t) - S_2(t)\right\| \leq (\mu + \eta_1) . \left\|S_1(t) - S_2(t)\right\|^{F-ABC} \mathbb{I}_{0,t}^{\varsigma_1,\varsigma_2}(1). \tag{24}$$

Simplifying (24), we obtain:

$$\left\|S_1(t) - S_2(t)\right\| - \left(\eta_1 + \mu\right) . \left\|S_1(t) - S_2(t)\right\|^{F-ABC} \mathbb{I}_{0,t}^{\varsigma_1,\varsigma_2}(1) \leq 0,$$

$$\left(1 - \left(\eta_1 + \mu\right)\right)\left\{^{F-ABC} \mathbb{I}_{0,t}^{\varsigma_1,\varsigma_2}(1)\right\} . \left\|S_1(t) - S_2(t)\right\| \leq 0,$$

$$(1 - (\mu + \eta_1))\left[\frac{\varsigma_1\varsigma_2}{\mathbb{ABC}(\varsigma_1)} \frac{\Gamma(\varsigma_2)}{\Gamma(\varsigma_1 + \varsigma_2)} + \frac{\varsigma_2(1 - \varsigma_1)\mathbb{Y}_1^{\varsigma_2-1}}{\mathbb{ABC}(\varsigma_1)}\right] . \left\|S_1(t) - S_2(t)\right\| \leq 0. \tag{25}$$

Thus, the term $(1 - (\mu + \eta_1))\left[\frac{\varsigma_1\varsigma_2}{\mathbb{ABC}(\varsigma_1)} \frac{\Gamma(\varsigma_2)}{\Gamma(\varsigma_1+\varsigma_2)} + \frac{\varsigma_2(1-\varsigma_1)\mathbb{Y}_1^{\varsigma_2-1}}{\mathbb{ABC}(\varsigma_1)}\right]$ cannot be equal to zero, therefore, we have

$$\|S_1(t) - S_2(t)\| = 0,$$

which gives

$$S_1(t) = S_2(t).$$

Hence, $S_1(t)$ and $S_2(t)$ are equal, which implies that the first compartment of model (12) has a unique solution.

Thus, we conclude that the operators $\mathbb{F}_1$, $\mathbb{F}_2$, $\mathbb{F}_3$, $\mathbb{F}_4$, and $\mathbb{F}_5$ are well-defined and satisfy the CPs. Therefore, the solution for our proposed WSNs SII$_1$PR model (12) is unique in the sense of the F-ABC derivative. $\qquad\square$

## 3.2 Stability analysis

In this section, we provide the results related to the stability analysis of our proposed model (12). To achieve this goal, we first establish the following results:

**Definition 7.** *The solution of proposed (12) is HUS in the FFI sense, if for $i \in \mathbb{N}_i^5$ there exists $\zeta_i > 0$ such that the following hold:*

$$\left| S(t) - \frac{\varsigma_1\varsigma_2}{\mathbb{ABC}(\varsigma_1)\Gamma(\varsigma_1)} \int_0^t (t-s)^{\varsigma_1-1} s^{*\varsigma_2-1} \mathbb{K}_1(S(t),t)ds \right.$$
$$\left. - \frac{\varsigma_2(1-\varsigma_1)t^{\varsigma_2-1}}{\mathbb{ABC}(\varsigma_1)} \mathbb{K}_1(S(t),t) \right| \leq \zeta_1,$$

$$\left| I(t) - \frac{\varsigma_1\varsigma_2}{\mathbb{ABC}(\varsigma_1)\Gamma(\varsigma_1)} \int_0^t (t-s)^{\varsigma_1-1} s^{*\varsigma_2-1} \mathbb{K}_2(I(t),t)ds \right.$$
$$\left. - \frac{\varsigma_2(1-\varsigma_1)t^{\varsigma_2-1}}{\mathbb{ABC}(\varsigma_1)} \mathbb{K}_2(I(t),t) \right| \leq \zeta_2,$$

$$\left| I_1(t) - \frac{\varsigma_1\varsigma_2}{\mathbb{ABC}(\varsigma_1)\Gamma(\varsigma_1)} \int_0^t (t-s)^{\varsigma_1-1} s^{*\varsigma_2-1} \mathbb{K}_3(I_1(t),t)ds \right.$$
$$\left. - \frac{\varsigma_2(1-\varsigma_1)t^{\varsigma_2-1}}{\mathbb{ABC}(\varsigma_1)} \mathbb{K}_3(I_1(t),t) \right| \leq \zeta_3,$$

$$\left| P(t) - \frac{\varsigma_1\varsigma_2}{\mathbb{ABC}(\varsigma_1)\Gamma(\varsigma_1)} \int_0^t (t-s)^{\varsigma_1-1} s^{*\varsigma_2-1} \mathbb{K}_4(P(t),t)ds \right.$$
$$\left. - \frac{\varsigma_2(1-\varsigma_1)t^{\varsigma_2-1}}{\mathbb{ABC}(\varsigma_1)} \mathbb{K}_4(P(t),t) \right| \leq \zeta_4,$$

$$\left| R(t) - \frac{\varsigma_1\varsigma_2}{\mathbb{ABC}(\varsigma_1)\Gamma(\varsigma_1)} \int_0^t (t-s)^{\varsigma_1-1} s^{*\varsigma_2-1} \mathbb{K}_5(R(t),t)ds \right.$$
$$\left. - \frac{\varsigma_2(1-\varsigma_1)t^{\varsigma_2-1}}{\mathbb{ABC}(\varsigma_1)} \mathbb{K}_4(R(t),t) \right| \leq \zeta_5, \tag{26}$$

*Let us assume that $\left(S^*(t), I^*(t), I_1^*(t), P^*(t), R^*(t)\right)$ be an approximate solution for (12) as:*

$$S^*(t) = \frac{\varsigma_1\varsigma_2}{\mathbb{ABC}(\varsigma_1)\Gamma(\varsigma_1)} \int_0^t (t-s)^{\varsigma_1-1} s^{*\varsigma_2-1} \mathbb{K}_1(S^*(t),t)ds + \frac{\varsigma_2(1-\varsigma_1)t^{\varsigma_2-1}}{\mathbb{ABC}(\varsigma_1)} \mathbb{K}_1(S^*(t),t),$$

$$I^*(t) = \frac{\varsigma_1\varsigma_2}{\mathbb{ABC}(\varsigma_1)\Gamma(\varsigma_1)} \int_0^t (t-s)^{\varsigma_1-1} s^{*\varsigma_2-1} \mathbb{K}_2(I^*(t),t)ds + \frac{\varsigma_2(1-\varsigma_1)t^{\varsigma_2-1}}{\mathbb{ABC}(\varsigma_1)} \mathbb{K}_2(I^*(t),t),$$

$$I_1^* t = \frac{\varsigma_1 \varsigma_2}{\mathbb{ABC}(\varsigma_1)\Gamma(\varsigma_1)} \int_0^t (t-s)^{\varsigma_1-1} s^{*\varsigma_2-1} \mathbb{K}_3(I_1^*(t),t) ds + \frac{\varsigma_2(1-\varsigma_1)t^{\varsigma_2-1}}{\mathbb{ABC}(\varsigma_1)} \mathbb{K}_3(I_1^*(t),t),$$

$$P^*(t) = \frac{\varsigma_1 \varsigma_2}{\mathbb{ABC}(\varsigma_1)\Gamma(\varsigma_1)} \int_0^t (t-s)^{\varsigma_1-1} s^{*\varsigma_2-1} \mathbb{K}_4(P^*(t),t) ds + \frac{\varsigma_2(1-\varsigma_1)t^{\varsigma_2-1}}{\mathbb{ABC}(\varsigma_1)} \mathbb{K}_4(P^*(t),t),$$

$$R^*(t) = \frac{\varsigma_1 \varsigma_2}{\mathbb{ABC}(\varsigma_1)\Gamma(\varsigma_1)} \int_0^t (t-s)^{\varsigma_1-1} s^{*\varsigma_2-1} \mathbb{K}_5(R^*(t),t) ds + \frac{\varsigma_2(1-\varsigma_1)t^{\varsigma_2-1}}{\mathbb{ABC}(\varsigma_1)} \mathbb{K}_5(R^*(t),t). \tag{27}$$

*Thus, the system of equations model ([12]), is Hyers-Ulam Stable, if*

$$|S - S^*| \leq \omega_1 \zeta_1,$$
$$|I - I^*| \leq \omega_2 \zeta_2,$$
$$|I_1 - I_1^*| \leq \omega_3 \zeta_3,$$
$$|P - P^*| \leq \omega_4 \zeta_4,$$
$$|R - R^*| \leq \omega_5 \zeta_5. \tag{28}$$

**Theorem 5.** *The solution of model ([12]) is Hyers-Ulam-Stable (HUS), if the system of inequalities ([28]) holds true.*

*Proof*: Let us take the first compartment of ([28]), we have

$$\begin{aligned}
\left|S - S^*\right| &= \left|\frac{\varsigma_1 \varsigma_2}{\mathbb{ABC}(\varsigma_1)\Gamma(\varsigma_1)} \int_0^t (t-s)^{\varsigma_1-1} s^{\varsigma_2-1} \mathbb{K}_1(S(t),t) ds + \frac{\varsigma_2(1-\varsigma_1)t^{\varsigma_2-1}}{\mathbb{ABC}(\varsigma_1)} \mathbb{K}_1(S(t),t)\right. \\
&\quad \left. - \left[\frac{\varsigma_1 \varsigma_2}{\mathbb{ABC}(\varsigma_1)\Gamma(\varsigma_1)} \int_0^t (t-s)^{\varsigma_1-1} s^{*\varsigma_2-1} \mathbb{K}_1(S^*(t),t) ds + \frac{\varsigma_2(1-\varsigma_1)t^{\varsigma_2-1}}{\mathbb{ABC}(\varsigma_1)} \mathbb{K}_1(S^*(t),t)\right]\right|, \\
&= \left|\frac{\varsigma_2(1-\varsigma_1)t^{\varsigma_2-1}}{\mathbb{ABC}(\varsigma_1)}\left[\mathbb{K}_1(S(t),t) - \mathbb{K}_1(S^*(t),t)\right]\right. \\
&\quad \left. + \frac{\varsigma_1 \varsigma_2}{\mathbb{ABC}(\varsigma_1)\Gamma(\varsigma_1)} \int_0^t (t-\psi)^{\varsigma_1-1} \psi^{\varsigma_2-1}\left[\mathbb{K}_1(S(t),t) - \mathbb{K}_1(S^*(t),t)\right]d\psi\right|, \\
&\leq \frac{\varsigma_2(1-\varsigma_1)t^{\varsigma_2-1}}{\mathbb{ABC}(\varsigma_1)} \psi_1 \left\|S(t) - S^*(t)\right\| \\
&\quad + \frac{\varsigma_1 \varsigma_2}{\mathbb{ABC}(\varsigma_1)\Gamma(\varsigma_1)} \int_0^t (t-\psi)^{\varsigma_1-1} \psi^{\varsigma_2-1} \psi_1 \left\|S(t) - S^*(t)\right\| d\psi, \\
&\leq \frac{\varsigma_2(1-\varsigma_1)t^{\varsigma_2-1}}{\mathbb{ABC}(\varsigma_1)} \psi_1 \left\|S(t) - S^*(t)\right\| \\
&\quad + \frac{\varsigma_1 \varsigma_2}{\mathbb{ABC}(\varsigma_1)\Gamma(\varsigma_1)} . \psi_1 \left\|S(t) - S^*(t)\right\| \int_0^t (t-\psi)^{\varsigma_1-1} \psi^{\varsigma_2-1} d\psi, \\
&\leq \frac{\varsigma_2(1-\varsigma_1)}{\mathbb{ABC}(\varsigma_1)} \psi_1 \left\|S(t) - S^*(t)\right\| \\
&\quad + \frac{\varsigma_1 \varsigma_2}{\mathbb{ABC}(\varsigma_1)\Gamma(\varsigma_1)} . \psi_1 \left\|S(t) - S^*(t)\right\| \mathbb{B}(\varsigma_1, \varsigma_2), \tag{29} \\
\left|S - S^*\right| &= \left[\frac{\varsigma_2(1-\varsigma_1)}{\mathbb{ABC}(\varsigma_1)} + \frac{\varsigma_1 \varsigma_2}{\mathbb{ABC}(\varsigma_1)\Gamma(\varsigma_1)} . \frac{\Gamma(\varsigma_1)\Gamma(\varsigma_2)}{\Gamma(\varsigma_1+\varsigma_2)}\right] . \psi_1 \left\|S(t) - S^*(t)\right\|, \\
&= \psi_1 . \left[\frac{\varsigma_2(1-\varsigma_1)}{\mathbb{ABC}(\varsigma_1)} + \frac{\varsigma_1 \varsigma_2}{\mathbb{ABC}(\varsigma_1)} . \frac{\Gamma(\varsigma_2)}{\Gamma(\varsigma_1+\varsigma_2)}\right] . \left\|S(t) - S^*(t)\right\|.
\end{aligned}$$

Now, let $\psi_1 = \zeta_1$ and $\left[\frac{\varsigma_2(1-\varsigma_1)}{\mathbb{ABC}(\varsigma_1)} + \frac{\varsigma_1\varsigma_2}{\mathbb{ABC}(\varsigma_1)} \cdot \frac{\Gamma(\varsigma_2)}{\Gamma(\varsigma_1+\varsigma_2)}\right] \cdot \left\|S(t) - S^*(t)\right\| = \omega_1$, thus

$$\left|S(t) - S^*(t)\right| \leq \omega_1\zeta_1,$$

which implies that the first compartment (12) is HUS. Similarly, we can obtain HUS results for the remaining compartments of the considered model. Therefore, we can say that the proposed system of Eq (12) is HUS. □

# 4 Control theory

In this section, we translate mathematical rigor into cyber-resilience, offering engineers a blueprint to outpace digital contagions. By anchoring worm dynamics to epidemiological principles like $R_0$, the framework acts as an early warning system: if $R_0 < 1$, networks can choke outbreaks before they metastasize. Sensitivity analysis reveals where defenses matter most, slowing data leaks $\beta$ and amplifying proactive patching $\rho$ become frontline tactics, akin to deploying a digital immune response. The fractional-order structure mirrors the messy reality of networks, where delays and legacy vulnerabilities linger, ensuring strategies adapt rather than crumble under uncertainty.

## 4.1 Equilibrium points

Here, mainly, we deal with the following equilibrium points.

### 4.1.1 Worm Free Equilibrium points (WFEP).
To find the WFEP of the system (12) algebraically, we set all the derivatives equal to zero by solving the equations of the system, and we get

$S(t) = 1 - \rho$, $\quad I(t) = 0$, $\quad I_1(t) = 0$, $\quad P(t) = \rho$, $\quad R(t) = 0$, with $S(t) + I(t) + P(t) + I_1(t) + R(t) = 1 = N(t)$.

Thus, the worm-free equilibrium is

$$\varphi_0 = (1 - \rho, \; 0, \; 0, \; \rho, \; 0).$$

### 4.1.2 Endemic Equilibrium (EE).
For finding the EE, we consider the following system of equations:

$$\begin{cases} (1 - \rho)\mu - \beta S^* I^* - \mu S^* = 0, \\ \dfrac{\beta S^* I^*}{N} - (\gamma_1 + \gamma + \mu + \delta_1)I^* = 0, \\ \delta_1 I^* - (\gamma_2 + \mu)I_1^* = 0, \\ \mu\rho - \mu P^* = 0, \\ \gamma_1 I^* - \mu R^* + \gamma_2 I_1^* = 0. \end{cases} \quad (30)$$

From the second equation, we have

$$\beta S^* - c = 0, \quad \text{where } c = \gamma_1 + \gamma + \mu + \delta_1 \implies S^* = \frac{c}{\beta}.$$

From the third equation, we have

$$I_1^* = \frac{\delta_1 I^*}{\gamma_2 + \mu}.$$

From the fourth equation, we have

$$P^* = \rho.$$

From the fifth equation, we have

$$R^* = \frac{\gamma_1 I^* + \gamma_2 I_1^*}{\mu} = \frac{\gamma_1 I^*}{\mu} + \frac{\gamma_2 \delta_1 I^*}{\mu(\gamma_2 + \mu)}.$$

Further, from the first compartment of model (12), we get

$$I^* = \frac{(1 - \rho)\mu}{c} - \frac{\mu}{\beta}.$$

## 4.2 Basic reproduction number $R_0$

The most critical threshold parameter concerning viral transmissibility is the basic reproduction number, typically stated as $R_0$.

**Definition 8.** *[39] $R_0$ is the average number of new infections that are transmitted by an infected individual throughout the entire period of infectiousness. If $R_0$ is greater than 1, the number of infected individuals will multiply rapidly and lead to an epidemic.*

The Basic Reproduction Number $R_0$ quantifies the inherent "spreadability" of a phenomenon in a system, answering: How widely will one source propagate its influence if left unchecked? In epidemiology, the principles of $R_0$ apply universally to measure the average downstream impact of a single unit (for example, a node) within a vulnerable network. If $R_0$ is greater than 1, the effect cascades exponentially, demanding mitigation to prevent system overload. If $R_0$ is less than 1, the influence disappears naturally. This metric helps design resilient systems by identifying critical thresholds for intervention, optimizing resource distribution, and predicting failure points. Although simplified, the logic of $R_0$ bridges theoretical models and real-world stability, offering a framework to balance efficiency and robustness in dynamic networks.

In the next-generation matrix (NGM) method, matrix F specifically describes the rate of new infections entering the system. In F, we only take I (primary infected nodes) due to biological and mathematical reasons considered within the framework of this model. $I_1$ is not epidemiologically related to new infections; rather, it is derived from I. The dynamics of the infected compartments I and $I_1$ are:

$$\frac{dI}{dt} = \beta S \frac{I}{N} - (\gamma_1 + \gamma + \mu + \delta_1) I, \tag{31}$$

$$\frac{dI_1}{dt} = \delta_1 I - (\gamma_2 + \mu) I_1. \tag{32}$$

Constructing **F** and **V**, the new infections (F) and transitions (P) matrices are:
**New infections matrix F**:

$$\mathbf{F} = \begin{bmatrix} \frac{\partial}{\partial I}(\beta S I) & \frac{\partial}{\partial I_1}(\beta S I) \\ \frac{\partial}{\partial I}(0) & \frac{\partial}{\partial I_1}(0) \end{bmatrix}_{\text{DFE}} = \begin{bmatrix} \beta(1 - \rho) & 0 \\ 0 & 0 \end{bmatrix}.$$

**Transitions matrix V:**

$$\mathbf{V} = \begin{bmatrix} \frac{\partial}{\partial I} [(\gamma_1 + \gamma + \mu + \delta_1)I] & \frac{\partial}{\partial I_1} [(\gamma_1 + \gamma + \mu + \delta_1)I]] \\ \frac{\partial}{\partial I} [-\delta_1 I + (\gamma_2 + \mu)I_1] & \frac{\partial}{\partial I_1} [-\delta_1 I + (\gamma_2 + \mu)I_1] \end{bmatrix}_{DFE} = \begin{bmatrix} \gamma_1 + \gamma + \mu + \delta_1 & 0 \\ -\delta_1 & \gamma_2 + \mu \end{bmatrix}.$$

$$\mathbf{V}^{-1} = \begin{bmatrix} \frac{1}{\gamma_1 + \gamma + \mu + \delta_1} & 0 \\ \frac{\delta_1}{(\gamma_1 + \gamma + \mu + \delta_1)(\gamma_2 + \mu)} & \frac{1}{\gamma_2 + \mu} \end{bmatrix}.$$

$$\mathbf{FV}^{-1} = \begin{bmatrix} \beta(1-\rho) & 0 \\ 0 & 0 \end{bmatrix} \begin{bmatrix} \frac{1}{\gamma_1 + \gamma + \mu + \delta_1} & 0 \\ \frac{\delta_1}{(\gamma_1 + \gamma + \mu + \delta_1)(\gamma_2 + \mu)} & \frac{1}{\gamma_2 + \mu} \end{bmatrix} = \begin{bmatrix} \frac{\beta(1-\rho)}{\gamma_1 + \gamma + \mu + \delta_1} & 0 \\ 0 & 0 \end{bmatrix}.$$

Spectral radius and $R_0$, the eigenvalues of $\mathbf{FV}^{-1}$ are

$$\lambda_1 = \frac{\beta(1-p)}{\gamma_1 + \gamma + \mu + \delta_1}, \quad \lambda_2 = 0.$$

Therefore, the basic reproduction number is the dominant eigenvalue of $F^{-1}V$ and hence:

$$\boxed{R_0 = \rho(\mathbf{FV}^{-1}) = \frac{\beta(1-\rho)}{\gamma_1 + \gamma + \mu + \delta_1}.}$$

**4.2.1 Sensitivity analysis of $R_0$.** The basic reproduction number $R_0$ is given by

$$R_0 = \frac{\beta(1-\rho)}{\gamma_1 + \gamma + \mu + \delta_1}.$$

To quantify the relative impact of parameters on $R_0$, we compute the **normalized sensitivity indices** (elasticity) using:

$$Y_\phi^{R_0} = \frac{\partial R_0}{\partial \phi} \times \frac{\phi}{R_0},$$

where $\phi$ is a parameter of interest. The following are the sensitivity indices for each parameter.

**Transmission Rate ($\beta$)**

$$Y_\beta^{R_0} = \frac{\partial R_0}{\partial \beta} \times \frac{\beta}{R_0} = \frac{(1-\rho)}{\gamma_1 + \gamma + \mu + \delta_1} \times \frac{\beta}{R_0} = +1.$$

Biologically this means that a 1% increase in $\beta$ increases $R_0$ by 1%.

**Protection Rate ($\rho$)**

$$Y_\rho^{R_0} = \frac{\partial R_0}{\partial \rho} \times \frac{\rho}{R_0} = \frac{-\beta}{\gamma_1 + \gamma + \mu + \delta_1} \times \frac{\rho}{R_0} = -\frac{\rho}{1-\rho}.$$

A 1% increase in $\rho$ decreases $R_0$ by $\frac{\rho}{1-\rho}\%$.

**Parameters in the denominator ($\gamma_1, \gamma, \mu, \delta_1$).** For any parameter $\phi \in \{\gamma_1, \gamma, \mu, \delta_1\}$:

$$Y_\phi^{R_0} = -\frac{\phi}{\gamma_1 + \gamma + \mu + \delta_1}.$$

For example: $Y_{\gamma_1}^{R_0} = -\frac{\gamma_1}{\gamma_1 + \gamma + \mu + \delta_1}$, $Y_\mu^{R_0} = -\frac{\mu}{\gamma_1 + \gamma + \mu + \delta_1}$.

The key observations for the sensitivity indices of $R_0$ from Table 4 are the following:

- The most sensitive parameter is $\beta$, with a direct 1:1 effect on $R_0$.
- The protection ($\rho$) is effective for reducing $R_0$, especially when $\rho$ is large.
- Parameters in the denominator ($\gamma_1, \gamma, \mu, \delta_1$) reduce $R_0$, but their impact depends on their relative magnitudes.

**4.2.2 Stability analysis on the bases of $R_0$.** Here, we present the result of stability and visualization of worm's transmission dynamics for the proposed model based on $R_0$, using the concepts of LaSalle's principle.

**Theorem 6.** *The worm-free equilibrium (WFE) is locally asymptotically stable when $R_0 < 1$ and unstable when $R_0 > 1$.*

*Proof*: We prove this via Jacobian analysis and the Lyapunov method. The Jacobian at WFE is:

$$J(\varphi_0) = \begin{pmatrix} -\mu & -\beta(1-\rho) & 0 & 0 & 0 \\ 0 & \beta(1-\rho)-c & 0 & 0 & 0 \\ 0 & \delta_1 & -d & 0 & 0 \\ 0 & 0 & 0 & -\mu & 0 \\ 0 & \gamma_1 & \gamma_2 & 0 & -\mu \end{pmatrix}, \quad \begin{aligned} c &= \gamma_1 + \gamma + \mu + \delta_1 \\ d &= \gamma_2 + \mu \end{aligned}$$

The eigenvalues are:

$$\lambda_1 = -\mu, \quad \lambda_2 = -d, \quad \lambda_3 = -\mu,$$
$$\lambda_4 = -\mu, \quad \lambda_5 = \beta(1-\rho) - c$$

All eigenvalues have negative real parts iff:

$$\beta(1-\rho) - c < 0 \iff R_0 = \frac{\beta(1-\rho)}{c} < 1$$

**Table 4. Normalized sensitivity indices of $R_0$ with increasing the parameter values.**

| Parameter | Sensitivity Index | Effect on $R_0$ |
|---|---|---|
| $\beta$ | $+1$ | Increases |
| $\rho$ | $-\frac{\rho}{1-\rho}$ | Decreases |
| $\gamma_1$ | $-\frac{\gamma_1}{\gamma_1+\gamma+\mu+\delta_1}$ | Decreases |
| $\gamma$ | $-\frac{\gamma}{\gamma_1+\gamma+\mu+\delta_1}$ | Decreases |
| $\mu$ | $-\frac{\mu}{\gamma_1+\gamma+\mu+\delta_1}$ | Decreases |
| $\delta_1$ | $-\frac{\delta_1}{\gamma_1+\gamma+\mu+\delta_1}$ | Decreases |

For $R_0 < 1$, define:

$$V(I, I_1) = I + \frac{\delta_1}{d} I_1$$

Its derivative along trajectories satisfies:

$$\dot{V} = \beta S \frac{I}{N} - cI + \frac{\delta_1}{d}(\delta_1 I - dI_1)$$

$$\leq \beta(1 - \rho)I - cI + \frac{\delta_1^2}{d}I - \delta_1 I_1$$

$$= (R_0 - 1)cI - \delta_1 I_1 \leq 0$$

Equality holds only when $I = I_1 = 0$. By LaSalle's invariance principle, WFE is globally attractive. When $R_0 > 1$, $\lambda_5 > 0$ makes $\varphi_0$ unstable. The transcritical bifurcation at $R_0 = 1$ is verified by computing:

$$a = \left.\frac{\partial^2 i}{\partial I^2}\right|_{\varphi_0} = \frac{2\beta\mu}{1 - \rho} > 0$$

□

The bifurcation is forward ($a > 0$), implying a stable endemic equilibrium for $R_0 > 1$.

To visualize how the spread of the worm depends on its contagiousness, the following Figs 2 and 3, show the transmission dynamics within the WSNs. They illustrate three distinct outbreak scenarios determined by the basic reproduction number, $R_0$. The value of $R_0$ critically influences whether the infection persists, grows, or dies out.

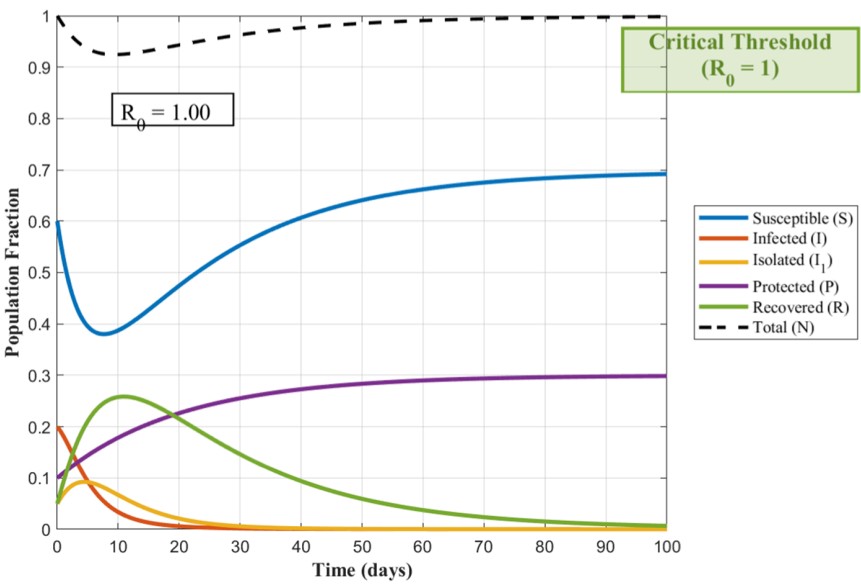

**Fig 2. WSN's Worm Transmission Dynamics for $R_0 = 1$.**

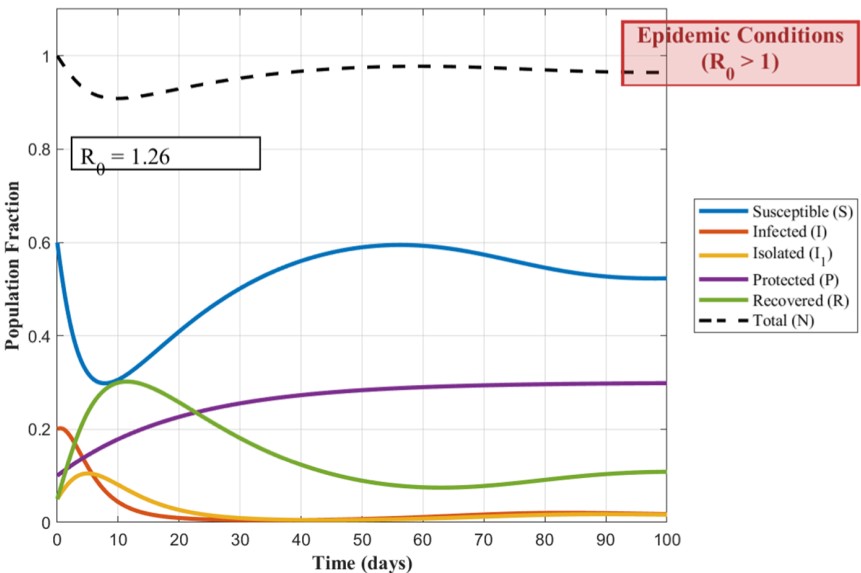

**Fig 3. WSNs' Worm Transmission Dynamics for $R_0 > 1$.**

Understanding how a worm spreads in a WSNs is crucial for developing defenses. The basic reproduction number $R_0$, acts as a key predictor for this spread. Fig 2 illustrates the critical threshold scenario where $R_0$ equals 1; here, each infected node transmits the worm to exactly one new node on average, leading to a persistent but non-expanding infection level. When $R_0$ exceeds 1, as shown in Fig 3, the outbreak grows exponentially because each infection causes

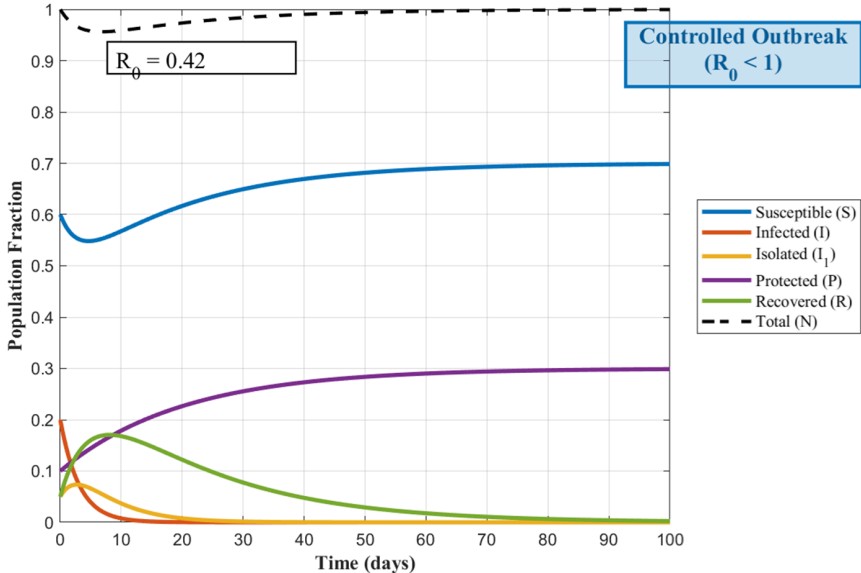

**Fig 4. WSNs Worm Transmission Dynamics for $R_0 < 1$.**

more than one subsequent infection, posing a significant threat to the network. Conversely, Fig 4 depicts the safer situation where $R_0$ is less than 1; the worm struggles to spread effectively, causing infections to quickly decline and eventually fizzle out. These dynamics clearly demonstrate how the value of $R_0$ determines the ultimate fate of a worm within the WSNs.

## 5 Newton's polynomial-based numerical scheme

This section details the numerical approximation of model (12) via Newton's polynomial approach, integrating interpolation techniques with Newton's method using Newton-form polynomials. These methodologies find extensive application across diverse fields such as physics, economics, and computer science.

For model (12), we delineate the numerical implementation scheme through the following steps:

As we know from the Atangana-Baleanu integral, the solution for the model (12) takes a specific form, which can be derived as follows:

$$\eta(t) = \eta(0) + \frac{(1-\varsigma_1)\,\varsigma_2\,t^{\varsigma_2-1}}{\mathbb{ABC}(\varsigma_1)}\mathbb{K}(t,\eta(t)) + \frac{\varsigma_1\varsigma_2}{\mathbb{ABC}(\varsigma_1)\Gamma(\varsigma_1)}\int_0^t (t-s)^{\varsigma_1-1}s^{\varsigma_2-1}\mathbb{K}(s,\eta(s))ds. \tag{33}$$

Substituting $t$ with $t_{j+1}$, in (32), we have

$$\eta(t_{j+1}) = \eta(0) + \frac{(1-\varsigma_1)\,\varsigma_2\,t_{j+1}^{\varsigma_2-1}}{\mathbb{ABC}(\varsigma_1)}\mathbb{K}\Big(t,\eta(t)\Big)$$
$$+ \frac{\varsigma_1\varsigma_2}{\mathbb{ABC}(\varsigma_1)\Gamma(\varsigma_1)}\int_0^{t_{j+1}} (t_{j+1}-\varphi)^{\varsigma_1-1}\varphi^{\varsigma_2-1}\mathbb{K}(\varphi,\eta(\varphi))d\varphi. \tag{34}$$

Now, applying the two step Lagrange polynomial to $\mathbb{K}(t,\eta(t))$, gives

$$\mathbb{K}(t,\eta(t)) = \frac{(\varphi - t_{\varphi-1})\mathbb{K}\Big(t_\varphi,\eta\big(t_{\varphi-1}\big)\Big)}{t_\varphi - t_{\varphi-1}} - \frac{(\varphi - t_\varphi)\mathbb{K}\Big(t_{\varphi-1},\eta\big(t_{\varphi-1}\big)\Big)}{t_\varphi - t_{\varphi-1}},$$
$$= \frac{\mathbb{K}\Big(t_\varphi,\eta\big(t_{\varphi-1}\big)\Big)(\varphi - t_{\varphi-1})}{t_\varphi - t_{\varphi-1}} - \frac{\mathbb{K}\Big(t_{\varphi-1},\eta\big(t_{\varphi-1}\big)\Big)(\varphi - t_\varphi)}{t_\varphi - t_{\varphi-1}},$$
$$= \frac{\mathbb{K}\Big(t_\varphi,\eta\big(t_{\varphi-1}\big)\Big)(\varphi - t_{\varphi-1})}{h} - \frac{\mathbb{K}\Big(t_{\varphi-1},\eta\big(t_{\varphi-1}\big)\Big)(\varphi - t_\varphi)}{h},$$

where the difference of $t_\varphi$ and $t_{\varphi-1}$ is represented by h.

$$\eta(t_{j+1}) = \eta(0) + \frac{(1-\varsigma_1)\varsigma_2\,t_{j+1}}{\mathbb{ABC}(\varsigma_1)}\mathbb{K}\Big(t,\eta(t)\Big)$$
$$+ \frac{\varsigma_1\varsigma_2}{\mathbb{ABC}(\varsigma_1)\Gamma(\varsigma_1)} \times \sum_{\varphi=1}^{n}\left[\left(\frac{\mathbb{K}(t_\varphi,\eta(t_\varphi))}{h}\int_0^{t_{j+1}}(t_{j+1}-\varphi)^{\varsigma_1-1}\,\varphi^{\varsigma_2-1}(\varphi - t_{\varphi-1})d\varphi\right)\right.$$
$$\left.- \frac{\mathbb{K}(t_{\varphi-1},\eta(t_{\varphi-1}))}{h}\int_0^{t_{j+1}}(t_{j+1}-\varphi)^{\varsigma_1-1}\varphi^{\varsigma_2-1}(\varphi - t_\varphi)d\varphi\right]. \tag{35}$$

Solving the integrals involving in Eq (34), we obtain

$$\eta(t_{j+1}) = \eta(0) + \frac{(1-\varsigma_1)\varsigma_2\, t_{j+1}}{\mathbb{ABC}(\varsigma_1)}\mathbb{K}\left(t,\eta(t)\right) + \frac{\varsigma_1\varsigma_2}{\mathbb{ABC}(\varsigma_1)\Gamma(\varsigma_1)}$$

$$\times \sum_{\varphi=1}^{n}\left[\left(\frac{\mathbb{K}(t_\varphi,\eta(t_\varphi))}{h}\left((t_{j+1})^{\varsigma_1+\varsigma_2}\frac{\Gamma(\varsigma_1)\Gamma(\varsigma_2+1)}{(\varsigma_1+\varsigma_2)\Gamma(\varsigma_1+\varsigma_2)} - \frac{t_{\varphi-1}}{t_{j+1}}\frac{\Gamma(\varsigma_1)\Gamma(\varsigma_2)}{\Gamma(\varsigma_1+\varsigma_2)}\right)\right)\right.$$

$$\left. - \frac{\mathbb{K}(t_{\varphi-1},\eta(t_{\varphi-1}))}{h}\left((t_{j+1})^{\varsigma_1+\varsigma_2}\frac{\Gamma(\varsigma_1)\Gamma(\varsigma_2+1)}{(\varsigma_1+\varsigma_2)\Gamma(\varsigma_1+\varsigma_2)} - \frac{t_\varphi}{t_{j+1}}\frac{\Gamma(\varsigma_1)\Gamma(\varsigma_2)}{\Gamma(\varsigma_1+\varsigma_2)}\right)\right].$$

After simplification, we get

$$\eta(t_{j+1}) = \eta(0) + \frac{(1-\varsigma_1)\varsigma_2\, t_{j+1}}{\mathbb{ABC}(\varsigma_1)}\mathbb{K}\left(t,\eta(t)\right) + \frac{\varsigma_1\varsigma_2}{\mathbb{ABC}(\varsigma_1)}$$

$$\times \sum_{\varphi=1}^{n}\left[\left(\frac{\mathbb{K}(t_\varphi,\eta(t_\varphi))}{h}\left((t_{j+1})^{\varsigma_1+\varsigma_2}\frac{\Gamma(\varsigma_2+1)}{(\varsigma_1+\varsigma_2)\Gamma(\varsigma_1+\varsigma_2)} - \frac{t_{\varphi-1}}{t_{j+1}}\frac{\Gamma(\varsigma_2)}{\Gamma(\varsigma_1+\varsigma_2)}\right)\right)\right.$$

$$\left. - \frac{\mathbb{K}(t_{\varphi-1},\eta(t_{\varphi-1}))}{h}\left((t_{j+1})^{\varsigma_1+\varsigma_2}\frac{\Gamma(\varsigma_2+1)}{(\varsigma_1+\varsigma_2)\Gamma(\varsigma_1+\varsigma_2)} - \frac{t_\varphi}{t_{j+1}}\frac{\Gamma(\varsigma_2)}{\Gamma(\varsigma_1+\varsigma_2)}\right)\right].$$

Replacing $\mathbb{K}\left(t_j,\eta\left(t_j\right)\right)$ *by* $\varsigma_2 t^{\varsigma_2-1}U\left(t,(t_j,\eta\left(t_j\right))\right)$, we get

$$\eta(t_{j+1}) = \eta(0) + \frac{(1-\varsigma_1)\varsigma_2^2\, t_{j+1}t^{\varsigma_2-1}}{\mathbb{ABC}(\varsigma_1)}U\left(t_j,\eta(t_j)\right)$$

$$+ \frac{\varsigma_1\varsigma_2^2 t^{\varsigma_2-1}\Gamma(\varsigma_2)}{h\mathbb{ABC}(\varsigma_1)\Gamma(\varsigma_1+\varsigma_2)}\times\sum_{\varphi=1}^{n}\left[\left(U(t_\varphi,\eta(t_\varphi))\left(\frac{(t_{j+1})^{\varsigma_1+\varsigma_2}\varsigma_2}{(\varsigma_1+\varsigma_2)} - \frac{t_{\varphi-1}}{t_{j+1}}\right)\right)\right.$$

$$\left. - U(t_{\varphi-1},\eta(t_{\varphi-1}))\left(\frac{(t_{j+1})^{\varsigma_1+\varsigma_2}\varsigma_2}{(\varsigma_1+\varsigma_2)} - \frac{t_\varphi}{t_{j+1}}\right)\right].$$

Now, applying the numerical scheme to model (12), we have

$$S(t_{j+1}) = S(0) + \frac{(1-\varsigma_1)\varsigma_2^2\, t_{j+1}t^{\varsigma_2-1}}{\mathbb{ABC}(\varsigma_1)}U\left(t_j,S(t_j)\right)$$

$$+ \frac{\varsigma_1\varsigma_2^2 t^{\varsigma_2-1}\Gamma(\varsigma_2)}{h\mathbb{ABC}(\varsigma_1)\Gamma(\varsigma_1+\varsigma_2)}\times\sum_{\varphi=1}^{n}\left[\left(U(t_\varphi,S(t_\varphi))\left(\frac{(t_{j+1})^{\varsigma_1+\varsigma_2}\varsigma_2}{(\varsigma_1+\varsigma_2)} - \frac{t_{\varphi-1}}{t_{j+1}}\right)\right)\right.$$

$$\left. - U(t_{\varphi-1},S(t_{\varphi-1}))\left(\frac{(t_{j+1})^{\varsigma_1+\varsigma_2}\varsigma_2}{(\varsigma_1+\varsigma_2)} - \frac{t_\varphi}{t_{j+1}}\right)\right],$$

$$I(t_{j+1}) = I(0) + \frac{(1-\varsigma_1)\varsigma_2^2\, t_{j+1}t^{\varsigma_2-1}}{\mathbb{ABC}(\varsigma_1)}U\left(t_j,I(t_j)\right)$$

$$+ \frac{\varsigma_1 \varsigma_2^2 t^{\varsigma_2-1}\Gamma(\varsigma_2)}{h\mathbb{ABC}(\varsigma_1)\Gamma(\varsigma_1+\varsigma_2)} \times \sum_{\varphi=1}^{n}\left[\left(U(t_\varphi, I(t_\varphi))\left(\frac{(t_{j+1})^{\varsigma_1+\varsigma_2}\varsigma_2}{(\varsigma_1+\varsigma_2)} - \frac{t_{\varphi-1}}{t_{j+1}}\right)\right)\right.$$

$$\left. - U(t_{\varphi-1}, I(t_{\varphi-1}))\left(\frac{(t_{j+1})^{\varsigma_1+\varsigma_2}\varsigma_2}{(\varsigma_1+\varsigma_2)} - \frac{t_\varphi}{t_{j+1}}\right)\right],$$

$$I_1(t_{j+1}) = I_1(0) + \frac{(1-\varsigma_1)\varsigma_2^2\, t_{j+1} t^{\varsigma_2-1}}{\mathbb{ABC}(\varsigma_1)} U\left(t_j, I_1(t_j)\right)$$

$$+ \frac{\varsigma_1 \varsigma_2^2 t^{\varsigma_2-1}\Gamma(\varsigma_2)}{h\mathbb{ABC}(\varsigma_1)\Gamma(\varsigma_1+\varsigma_2)} \times \sum_{\varphi=1}^{n}\left[\left(U(t_\varphi, I_1(t_\varphi))\left(\frac{(t_{j+1})^{\varsigma_1+\varsigma_2}\varsigma_2}{(\varsigma_1+\varsigma_2)} - \frac{t_{\varphi-1}}{t_{j+1}}\right)\right)\right.$$

$$\left. - U(t_{\varphi-1}, I_1(t_{\varphi-1}))\left(\frac{(t_{j+1})^{\varsigma_1+\varsigma_2}\varsigma_2}{(\varsigma_1+\varsigma_2)} - \frac{t_\varphi}{t_{j+1}}\right)\right],$$

$$P(t_{j+1}) = P(0) + \frac{(1-\varsigma_1)\varsigma_2^2\, t_{j+1} t^{\varsigma_2-1}}{\mathbb{ABC}(\varsigma_1)} U\left(t_j, P(t_j)\right)$$

$$+ \frac{\varsigma_1 \varsigma_2^2 t^{\varsigma_2-1}\Gamma(\varsigma_2)}{h\mathbb{ABC}(\varsigma_1)\Gamma(\varsigma_1+\varsigma_2)} \times \sum_{\varphi=1}^{n}\left[\left(U(t_\varphi, P(t_\varphi))\left(\frac{(t_{j+1})^{\varsigma_1+\varsigma_2}\varsigma_2}{(\varsigma_1+\varsigma_2)} - \frac{t_{\varphi-1}}{t_{j+1}}\right)\right)\right.$$

$$\left. - U(t_{\varphi-1}, P(t_{\varphi-1}))\left(\frac{(t_{j+1})^{\varsigma_1+\varsigma_2}\varsigma_2}{(\varsigma_1+\varsigma_2)} - \frac{t_\varphi}{t_{j+1}}\right)\right],$$

$$R(t_{j+1}) = R(0) + \frac{(1-\varsigma_1)\varsigma_2^2\, t_{j+1} t^{\varsigma_2-1}}{\mathbb{ABC}(\varsigma_1)} U\left(t_j, R(t_j)\right)$$

$$+ \frac{\varsigma_1 \varsigma_2^2 t^{\varsigma_2-1}\Gamma(\varsigma_2)}{h\mathbb{ABC}(\varsigma_1)\Gamma(\varsigma_1+\varsigma_2)} \times \sum_{\varphi=1}^{n}\left[\left(U(t_\varphi, R(t_\varphi))\left(\frac{(t_{j+1})^{\varsigma_1+\varsigma_2}\varsigma_2}{(\varsigma_1+\varsigma_2)} - \frac{t_{\varphi-1}}{t_{j+1}}\right)\right)\right.$$

$$\left. - U(t_{\varphi-1}, R(t_{\varphi-1}))\left(\frac{(t_{j+1})^{\varsigma_1+\varsigma_2}\varsigma_2}{(\varsigma_1+\varsigma_2)} - \frac{t_\varphi}{t_{j+1}}\right)\right].$$

## 6 Numerical simulations

This section provides graphical visualizations and a concise discussion of the approximate solutions obtained for our proposed generalized fractional-order WSNs model.

### 6.1 Graphical analysis and discussion

We have used MATLAB to analyze and generate the graphs given below. The parameter values were obtained in order to get a graphical representation of the dynamics of our proposed model. The model parameters consist of $\beta, \rho, \mu, \gamma, \gamma_1, \delta_1$ and $\gamma_2$, each with specific values and corresponding units. Let us assume the following initial conditions:

$$S_0 = 1000, \quad I_0 = 10, \quad I_{10} = 0, \quad P_0 = 0, \quad R_0 = 0.$$

Thus, the data presented here lay the foundation for developing the model we are examining. The simulation results of the $SII_1PR$ model are shown in Figs 5–20, illustrating the dynamical representation of the time evolution of each compartment. These figures demonstrate to exhibit the dynamics of the system under different parameter values. For example, as the parameters $\varsigma_1$ and $\varsigma_2$ increase, there is a notable reduction in the susceptible population $S(t)$ shown in Fig 7. The behavioral pattern of $S(t)$, $P(t)$, and $R(t)$ in Figs 5, 16–19 also substantiates the theoretical findings, affirming the boundedness, convergence, and long-term influence of protection mechanisms. We see that the infected nodes $I(t)$ in Fig 10 first rise and then fall as a result of the impact of isolation and recovery parameters. In addition, the infected $I(t)$ presented and isolated $I_1(t)$ illustrated in Figs 10 and 13 exhibit sharp oscillations, reaching their peak before rapidly declining. This behavior suggests an initial aggressive infection that can be mitigated through strategic interventions. Likewise, the isolated class $I_1(t)$ in Fig 13 presents an increase followed by stabilization, establishing the efficacy of the containment strategy. Such graphical interpretations justify the model's ability to catch worm propagation patterns in WSNs. These dynamics emphasize the model's sensitivity to parameter changes and demonstrate the impact of $\varsigma_1$ and $\varsigma_2$ on infection control in wireless sensor networks. The Figs 21, 22, 26, 27, 28 and 29 reveal how cybersecurity "knobs" shape worm outbreaks in WSNs. Infected nodes Fig 21 surge faster with higher transmission rates ($\beta$), but flatten with stronger protection ($\rho$). Isolated nodes in Fig 22 peak earlier when isolation is swift ($\delta_1$), while faster recovery ($\gamma_2$) shortens isolation periods, but node death ($\mu$) permanently reduces network capacity. Protected nodes Figs 23–25 grow rapidly with preemptive hardening ($\rho$), though new nodes ($\mu$) introduce vulnerabilities. Recovered nodes Fig 26 thrive with efficient remediation ($\gamma_1, \gamma_2$), but node loss ($\mu$) forces costly replacements. Finally, combined dynamics Figs 27–29 show fractional parameters $\varsigma_1, \varsigma_2$ capturing real world delays: infections ripple through $S \to I \to I_1 \to R$, while protection $P$ curbs susceptibility. Together, they prove that early isolation and proactive patching are critical and that fractional calculus models real world quirks like "memory effects" in nodes.

The following are the graphs for different values of parameters.

In general, the analysis indicates that increasing these parameters leads to the following:

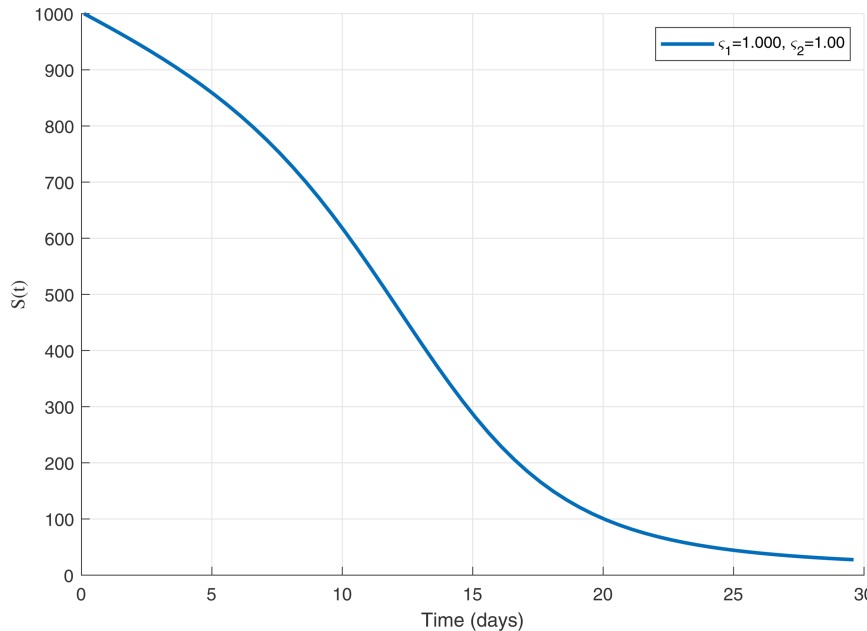

**Fig 5**. The dynamics of S(t) with $\varsigma_1, \varsigma_2 = 1$.

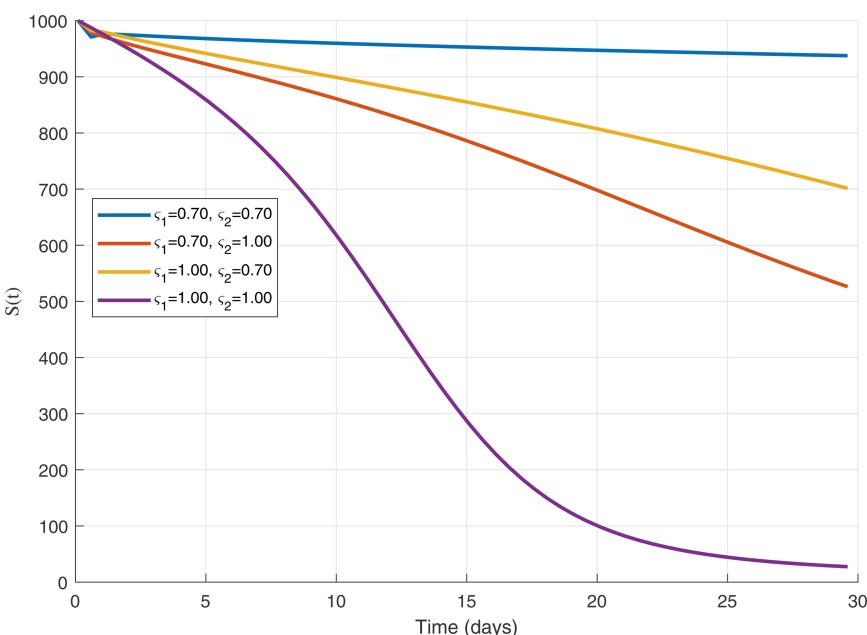

**Fig 6**. The dynamics of S(t) with variations in fractional orders $\varsigma_1$ and $\varsigma_2$ with each others.

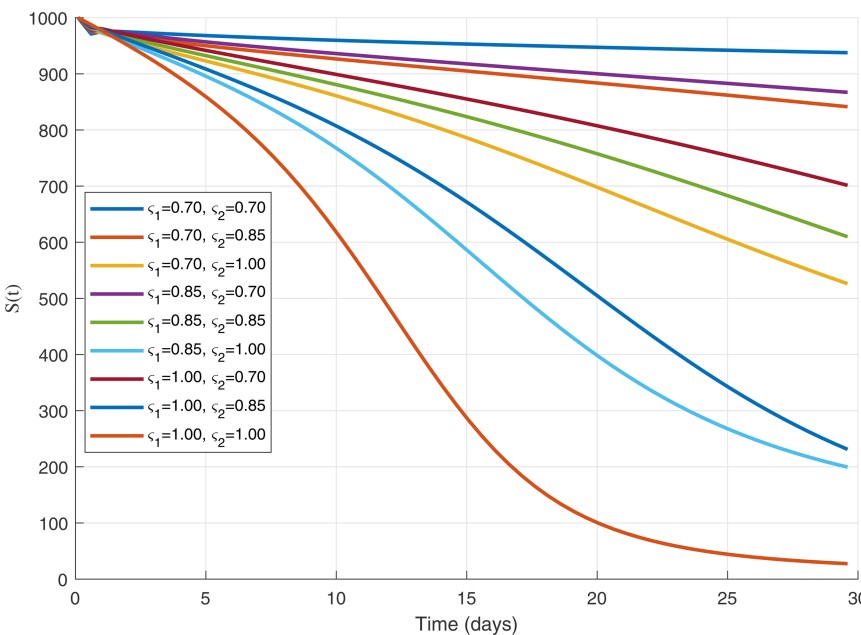

**Fig 7**. The S(t) population dynamics for multiple values of $\varsigma_1$ and $\varsigma_2$.

- A faster decline in the susceptible population $S(t)$ shown in Fig 7.
- Rapid fluctuations illustrated in infection $I(t)$ presented in Fig 10 and isolated $I_1(t)$ in Fig 13.
- A steady rise in protection $P(t)$ shown in Fig 16, and recovered nodes $R(t)$ presented in Fig 19.

**Fig 8. The dynamic of I(t) for $\varsigma_1, \varsigma_2 = 1.00$.**

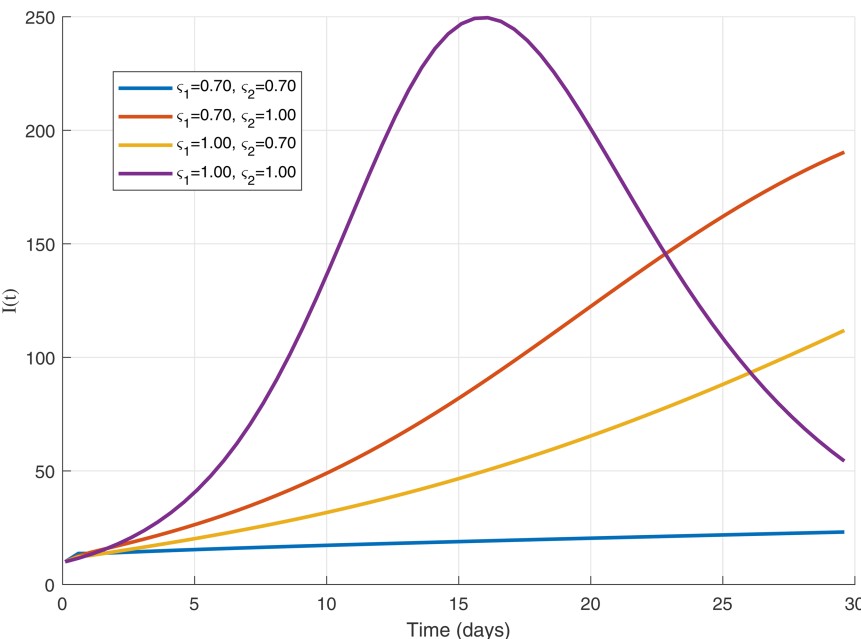

**Fig 9. The dynamics of I(t) with variations in $\varsigma_1$ and $\varsigma_2$ with each others.**

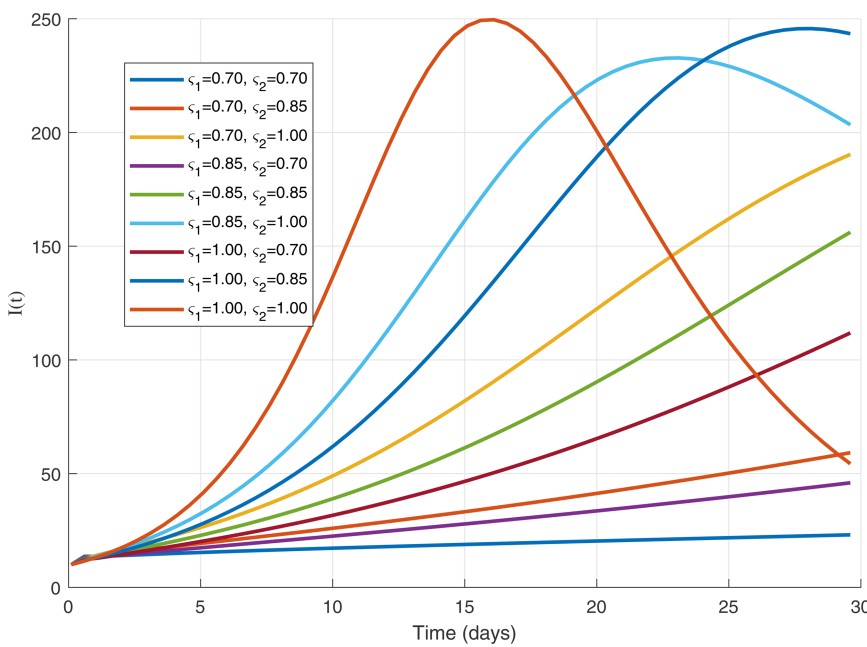

**Fig 10**. The I(t) dynamics for multiple values of $\varsigma_1$ and $\varsigma_2$.

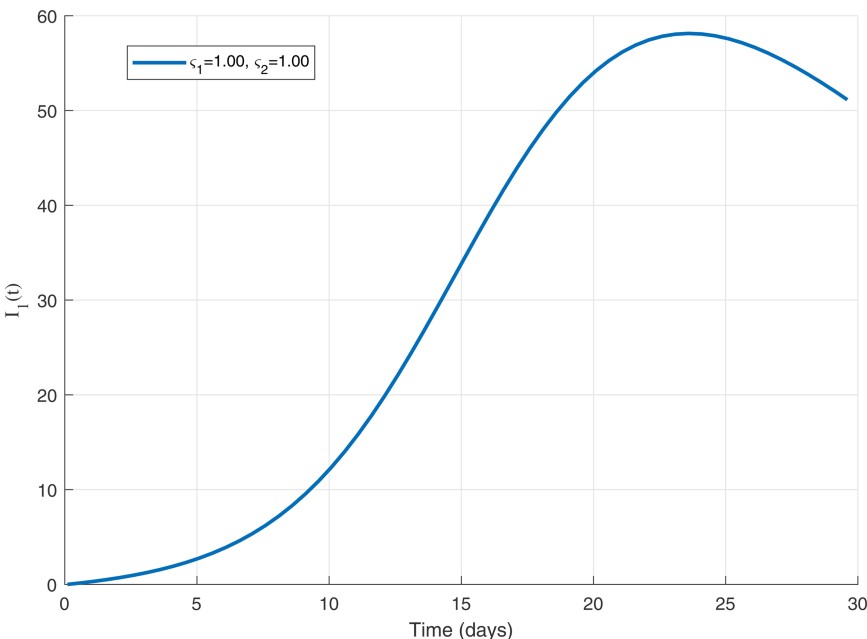

**Fig 11**. The dynamics of $I_1(t)$ for $\varsigma_1, \varsigma_2 = 1.0$.

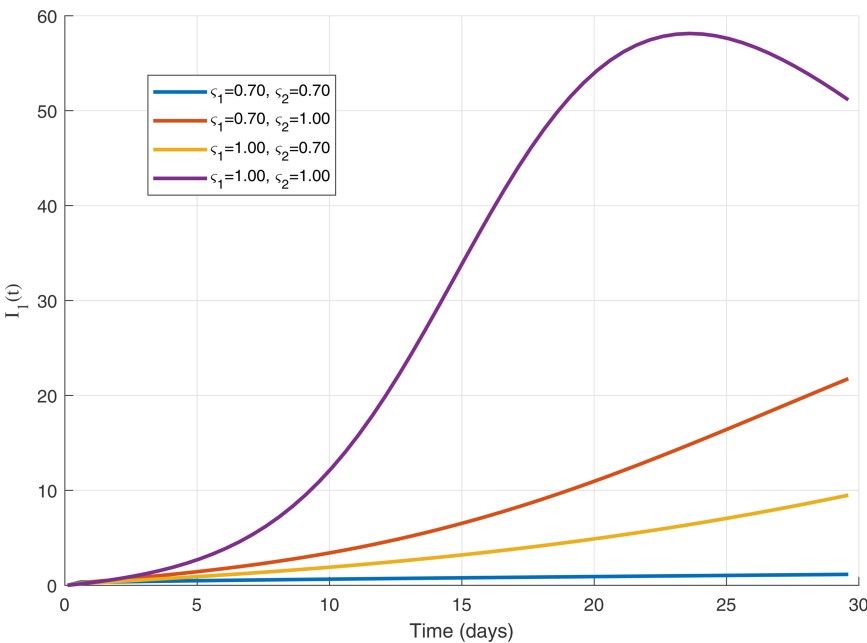

**Fig 12. The dynamics of $I_1(t)$ for variations of $\varsigma_1$ and $\varsigma_2$ with each others.**

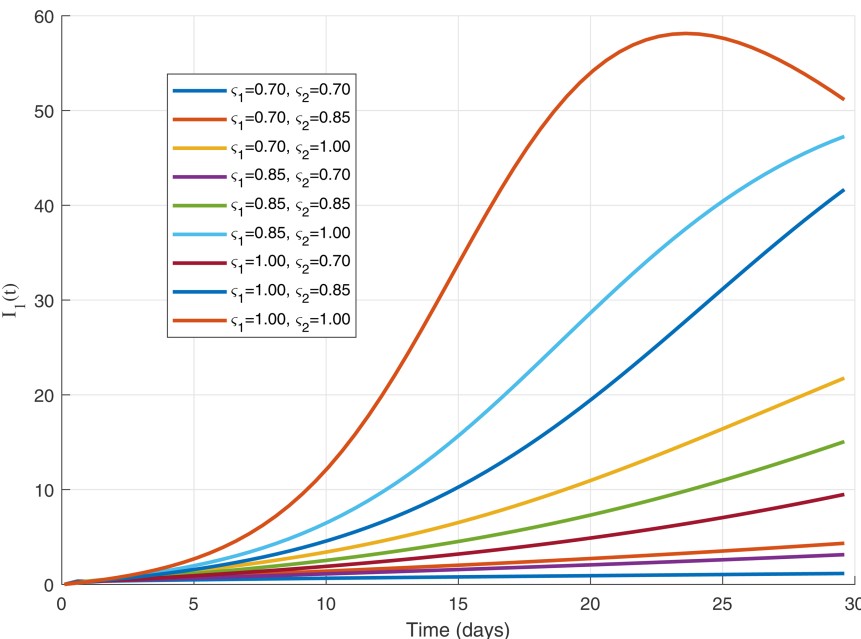

**Fig 13. The dynamics of $I_1(t)$ for multiple values of $\varsigma_1$ and $\varsigma_2$.**

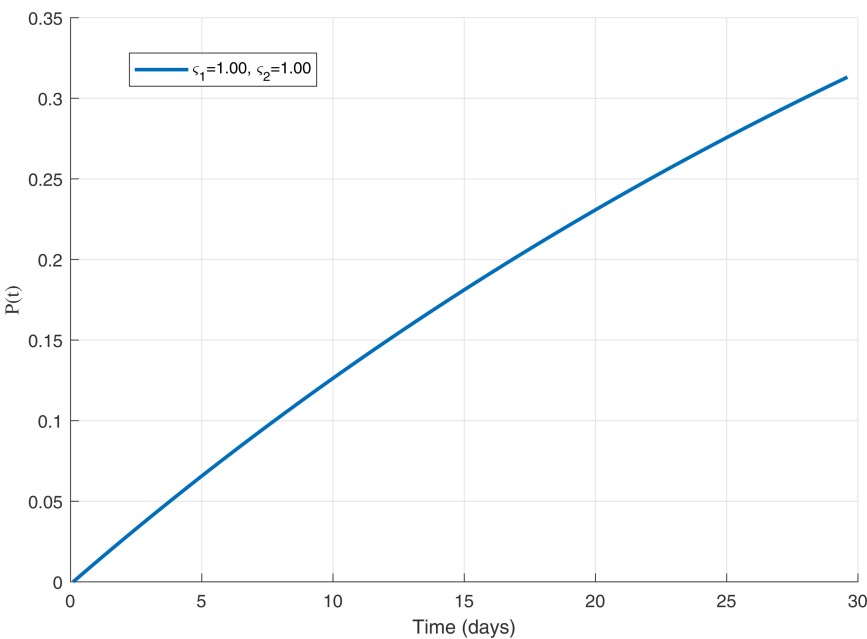

**Fig 14. Plot for Protected population P(t) with $\varsigma_1, \varsigma_2 = 1.00$.**

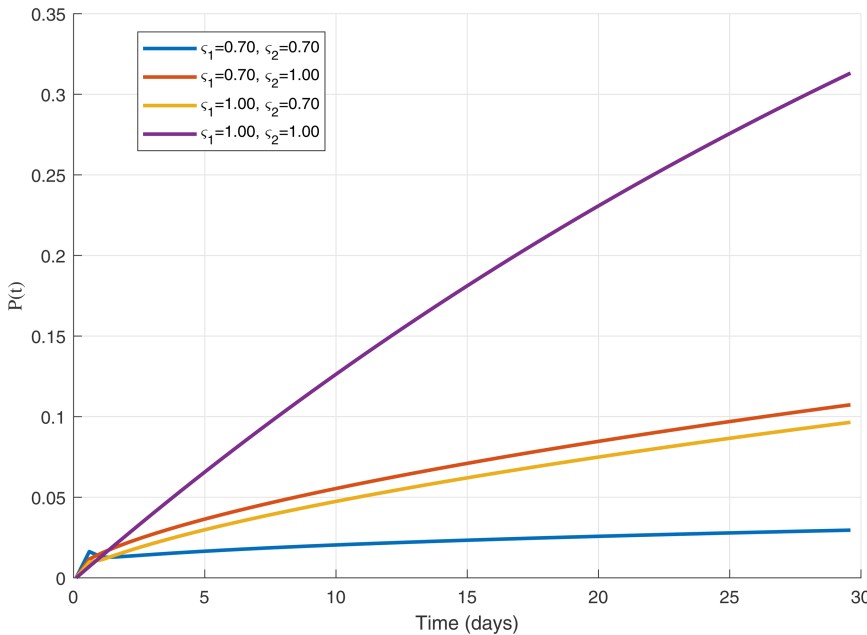

**Fig 15. The dynamics of P(t) with variations in values of $\varsigma_1$ and $\varsigma_2$ each others.**

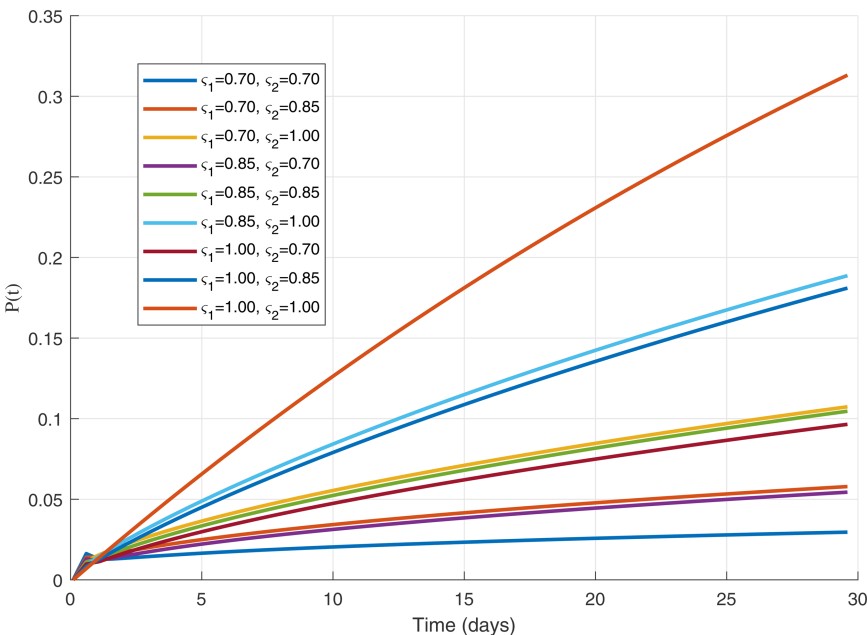

**Fig 16**. The P(t) populations for multiple values of $\varsigma_1$ and $\varsigma_2$.

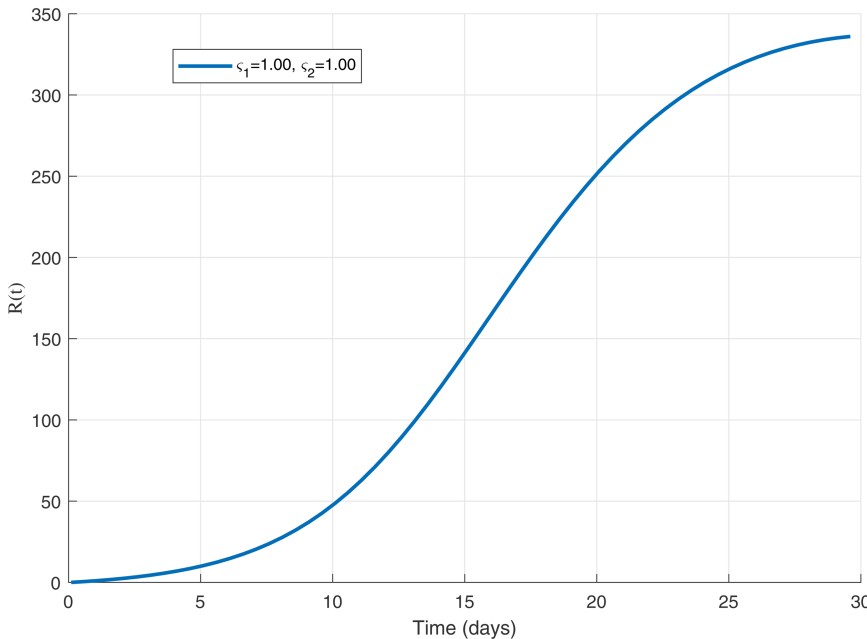

**Fig 17**. The dynamics of R(t) for $\varsigma_1, \varsigma_2 = 1.00$.

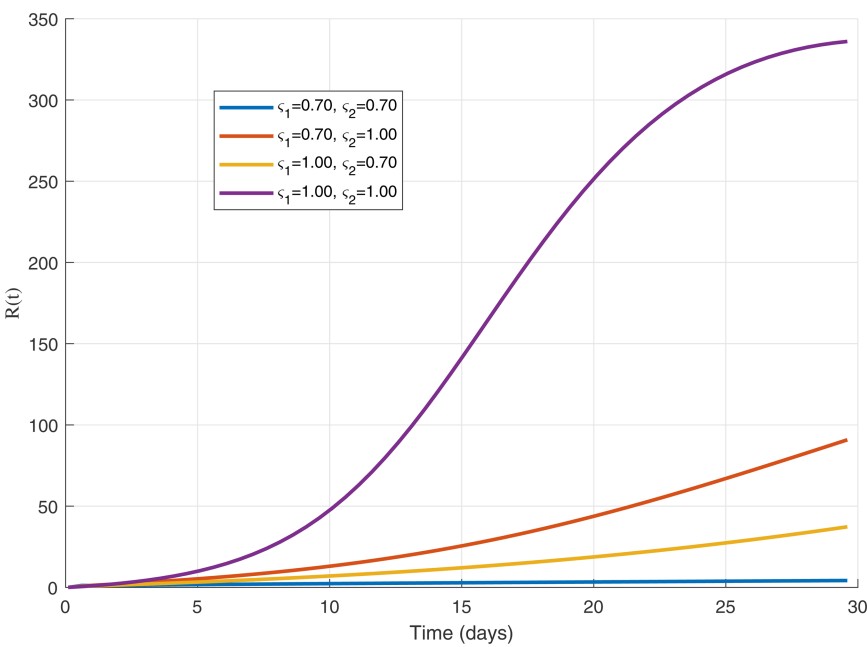

**Fig 18**. The dynamics of R(t) for variations in values of $\varsigma_1$ and $\varsigma_2$ each others.

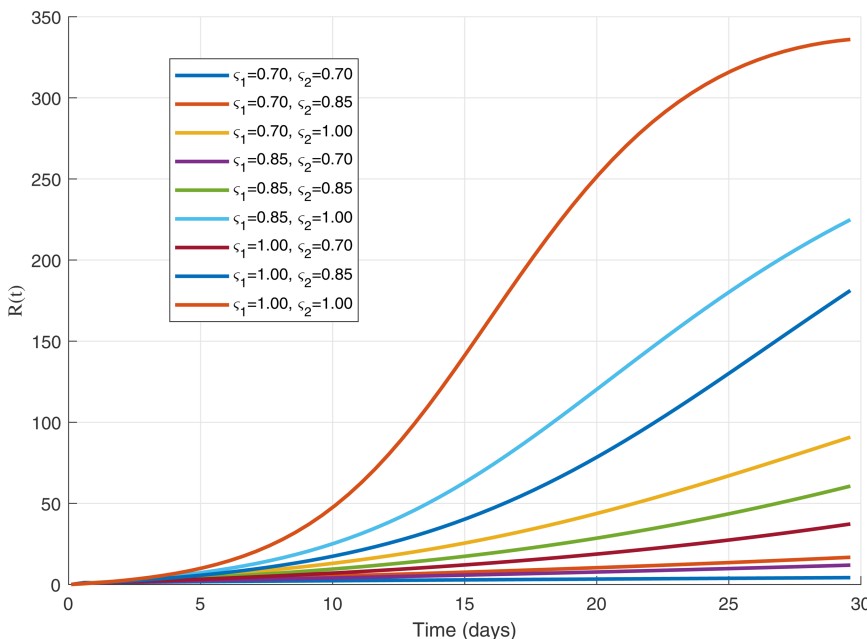

**Fig 19**. The dynamics of R(t) for multiple values of $\varsigma_1$, and $\varsigma_2$.

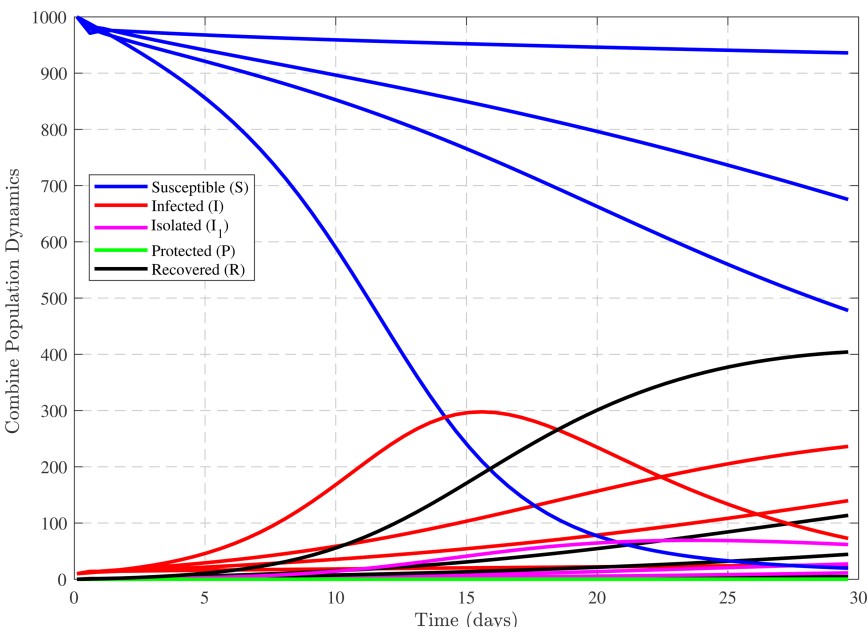

**Fig 20**. **Combining plot all compartments of SII$_1$PR model for multiple values of $\varsigma_1$ and $\varsigma_2$.**

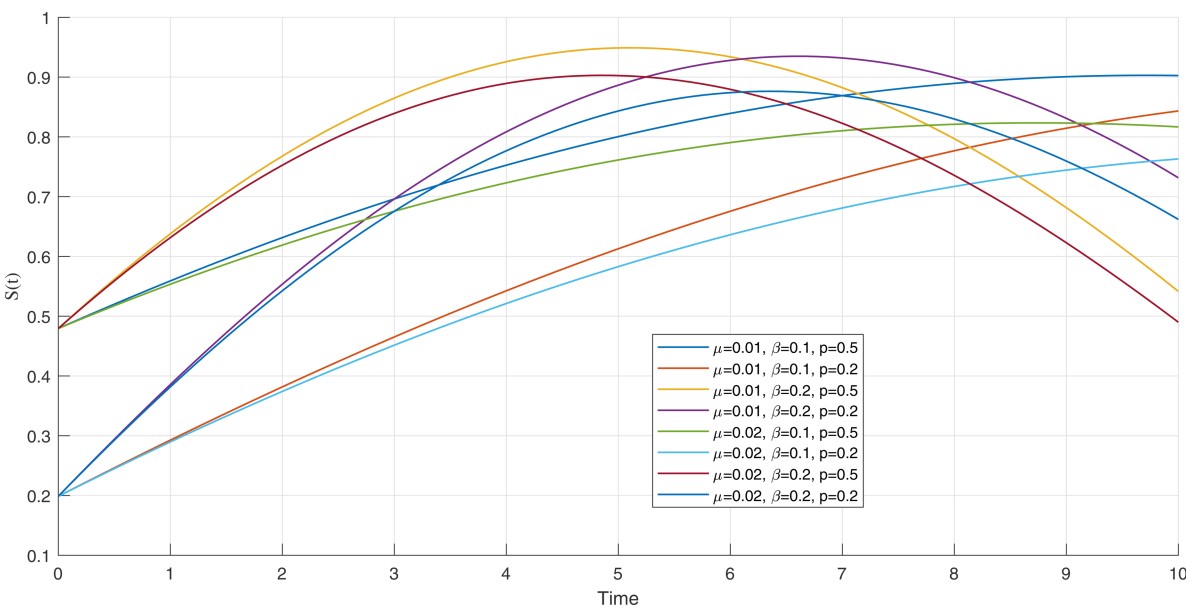

**Fig 21**. **Susceptible population dynamics with different values of parameters.**

This underscores the effectiveness of strategic interventions in controlling the spread of worms in WSNs.

## 6.2 Comparative study

To evaluate the effectiveness of the proposed numerical scheme, we conducted comparative simulations with classical methods:

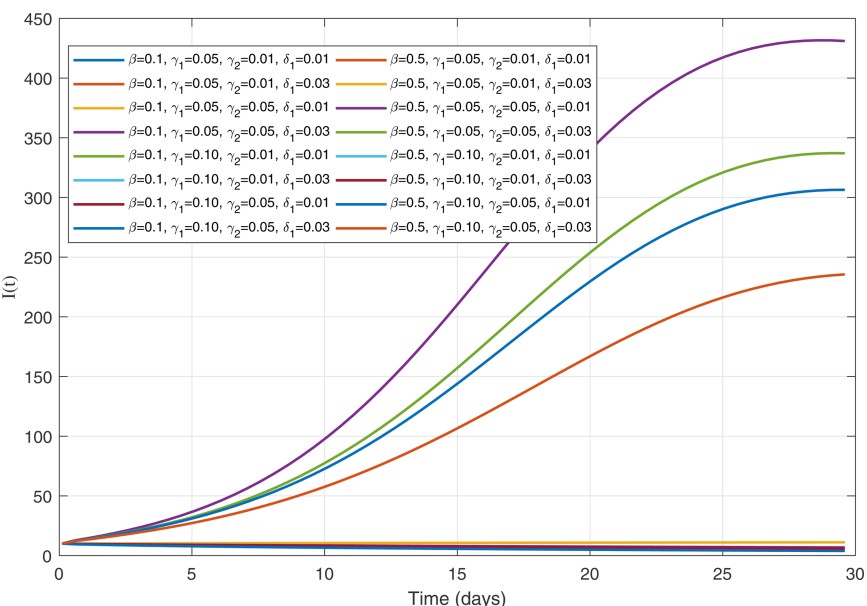

**Fig 22. Infected population dynamics with different values of parameters.**

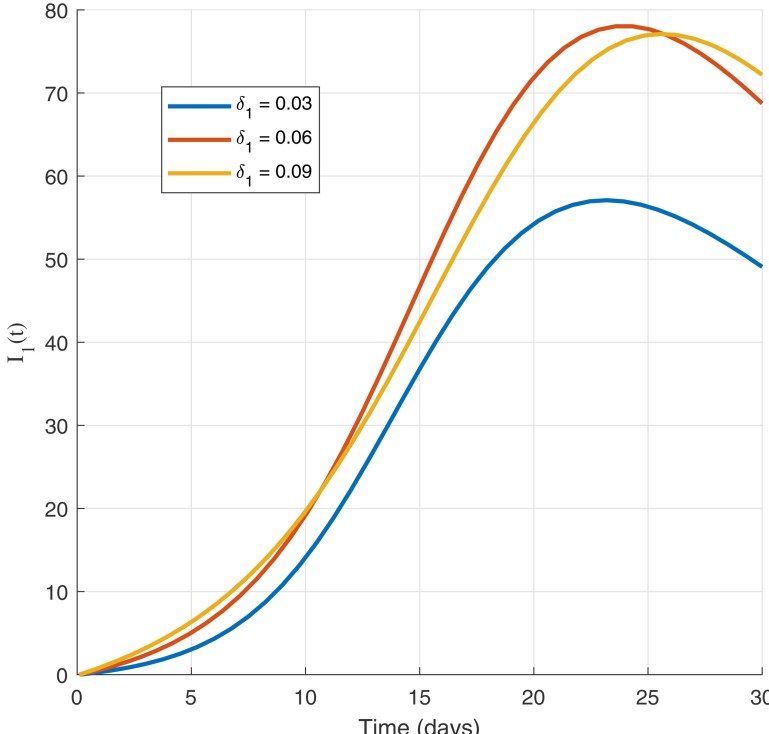

**Fig 23. Isolated population dynamics for different values of $\delta_1$.**

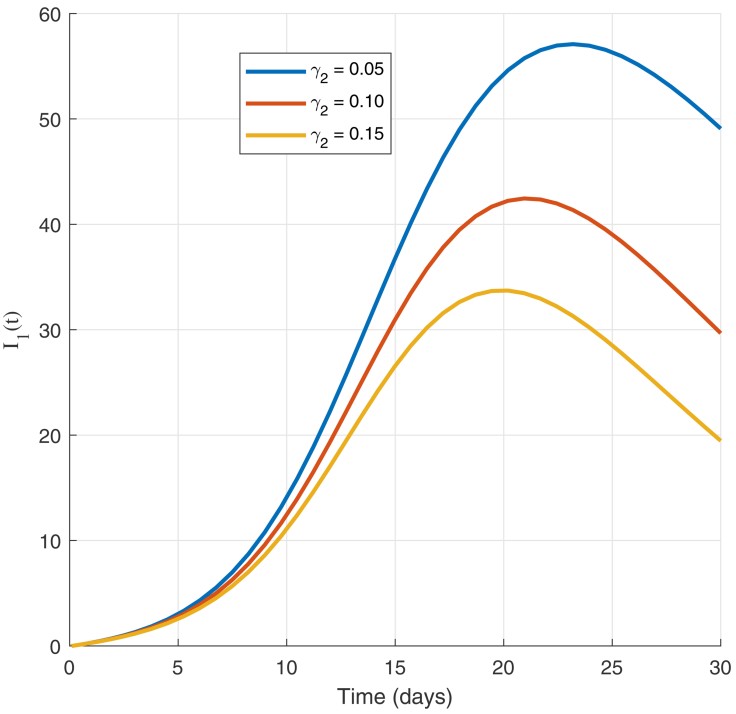

**Fig 24. Isolated population dynamics for different values of $\gamma_2$.**

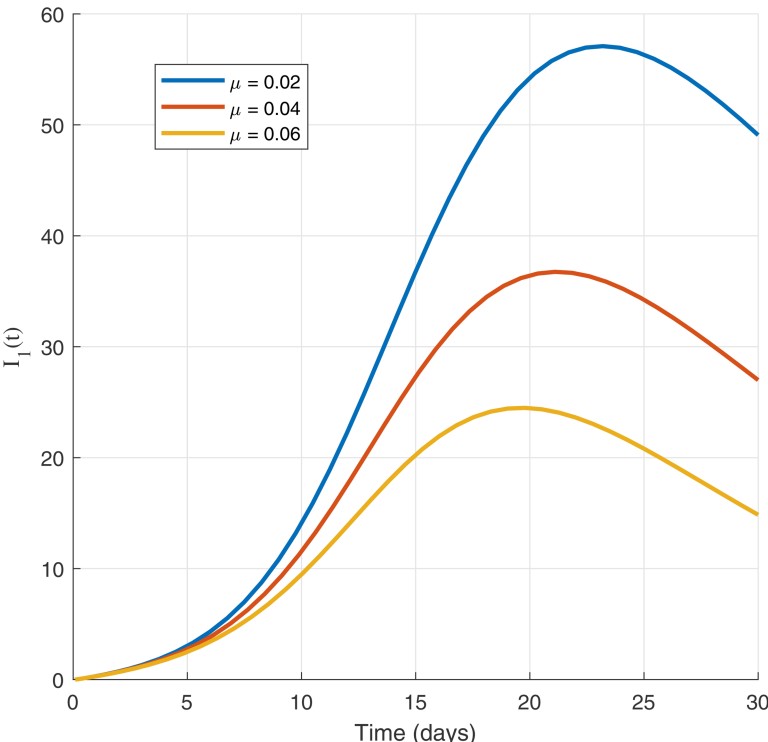

**Fig 25. Isolated population dynamics for different values of $\mu$.**

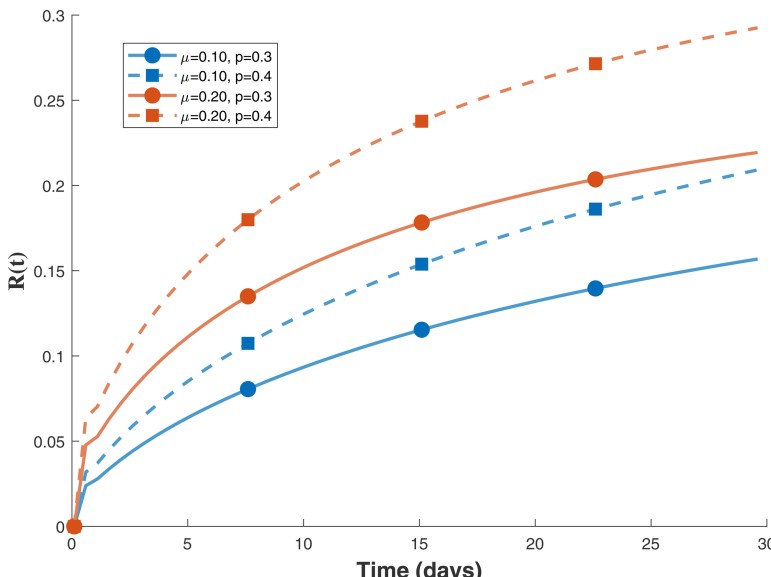

**Fig 26. Protected population dynamics with different values of parameters.**

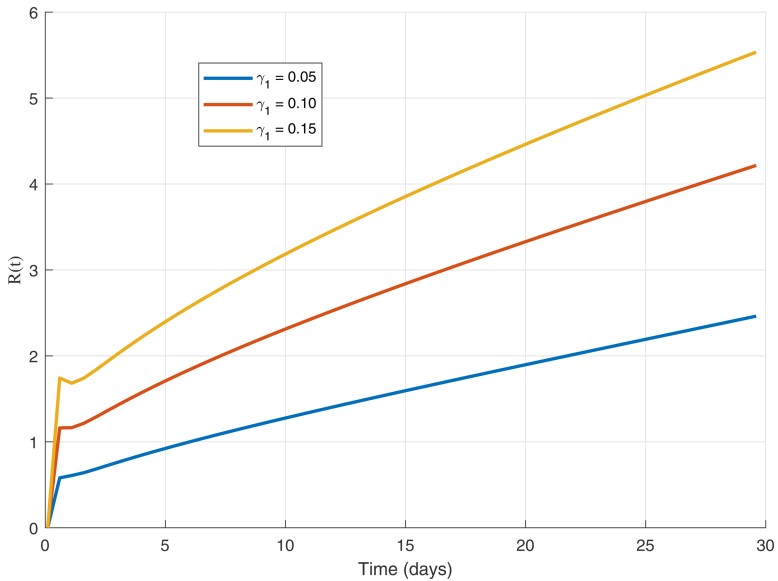

**Fig 27. Recovered population dynamics for different values of $\gamma_1$.**

- 4th-order Runge-Kutta (RK-4) and modified Euler method (EM)
- Conventional integer-order case ($\varsigma_1 = \varsigma_2 = 1$)
- MATLAB R2015a with fixed step size $h = 0.5$
- CPU time comparison for the susceptible compartment $S(\tau)$

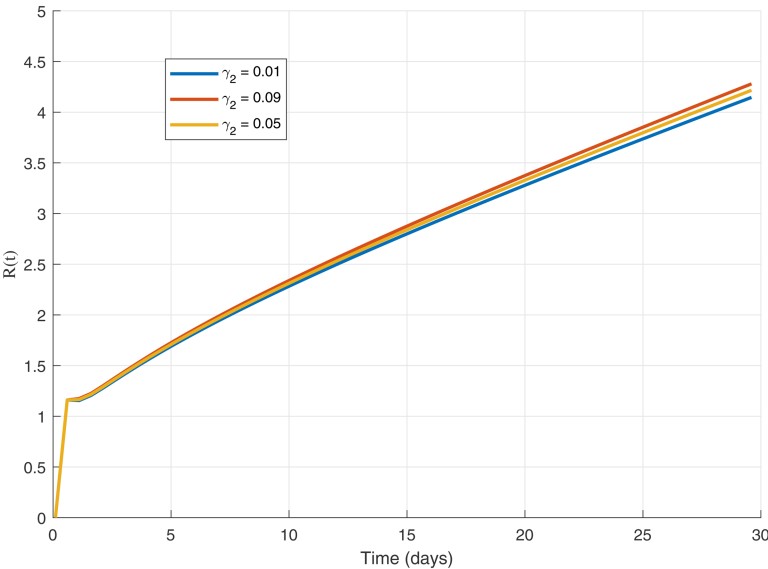

**Fig 28. Recovered population dynamics for different values of $\gamma_2$.**

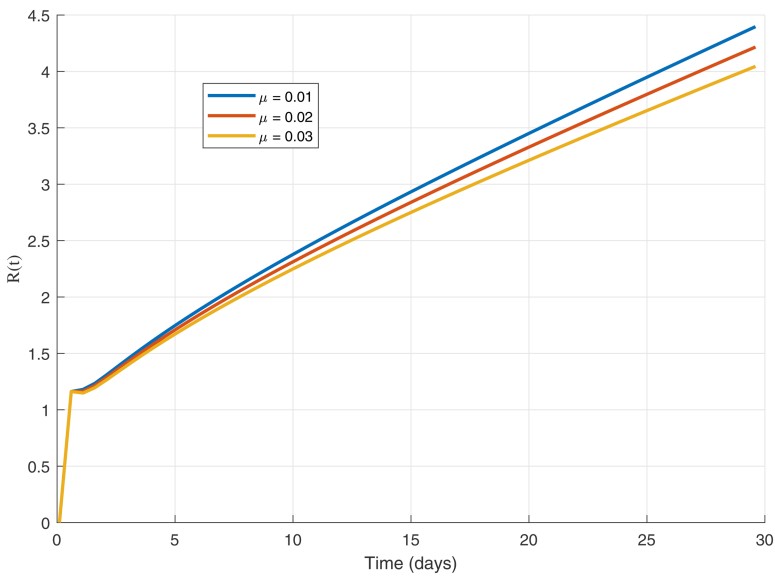

**Fig 29. Recovered population dynamics for different values of $\mu$.**

We compared the computational efficiency of the numerical schemes by measuring their CPU times, as shown in Table 1. The results demonstrate that our proposed method is more reliable than both RK-4 and EM approaches. Moreover, the NP scheme maintains its effectiveness for fractional-order cases ($\varsigma_1, \varsigma_2 < 1$), where traditional methods like RK-4 fail to apply. This adaptability makes the NP method particularly valuable for modeling real-world phenomena exhibiting memory effects and non-local behavior, such as contaminant transport in aquatic systems.

**Table 5**. CPU time comparison for extended simulation intervals using NP, RK4, and ME methods.

| $N = \frac{t}{h}$ | Time | RK4 CPUTime(s) | ME CPUTime(s) | NP CPUTime(s) |
|---|---|---|---|---|
| 100 | 50 | 0.129 | 0.114 | 0.092 |
| 200 | 100 | 0.251 | 0.228 | 0.188 |
| 300 | 150 | 0.372 | 0.340 | 0.284 |
| 400 | 200 | 0.503 | 0.451 | 0.369 |

## 7 Conclusions

This study presented an advanced mathematical model for analyzing worm-spreading dynamics in WSNs.

- This work extends the traditional (SIPR) model by incorporating an isolated compartment ($I_1$), introducing new parameters $\delta_1$ (isolation rate) and $\gamma_2$ (recovery rate of $I_1$), and utilizing FFD in the ABC sense. Incorporating an isolated compartment into the model improves its alignment with real-world scenarios by acknowledging nodes that are purposefully isolated to reduce the spread of worms. The ABC fractional derivative is employed to represent memory-influenced behavior and network-wide interactions, which are critical characteristics of malware propagation.
- By studying the $SII_1PR$ compartmental framework, researchers can identify how the parameters $\varsigma_1$ and $\varsigma_2$, the measurable distinct and measure functions, are used in managing worm outbreaks within WSNs.
- This analysis helps quantify the efficacy of containment strategies and track the restoration of compromised nodes to normal operation, offering insights into both preventive and reactive measures.
- These parameters, representing transmission dynamics and control interventions, respectively have a significant impact across all compartments. Fixing $\varsigma_1$ and $\varsigma_2$ values produces a typical epidemic curve, whereas varying these parameters scales outbreak severity and duration. Higher values of $\varsigma_1$ accelerate transmission, while increasing $\varsigma_2$ reduces infection peaks, indicating its role in mitigation measures.
- The Isolation patterns inversely reflect infection rates, with $\varsigma_1$ and $\varsigma_2$ influencing the timeliness and severity of isolation measures. The combined patterns $SII_1PR$ show that the balanced modifications to $\varsigma_1$ and $\varsigma_2$, such as maximizing transmission suppression while enhancing vaccination efforts, can effectively flatten the infection curves.

## Future works

This study established a theoretical foundation for modeling worm propagation in WSNs using fractal-fractional operators. Subsequent research should focus on validating the model empirically through test bed implementations with commercial sensor nodes, examining real-world constraints like dynamic topologies and intermittent connectivity. Further extensions could incorporate multi-vector threat scenarios (e.g., simultaneous malware strains) and energy-aware mitigation strategies that optimize security-energy tradeoffs. Developing machine learning frameworks for real-time parameter calibration and exploring hardware-level isolation mechanisms would enhance practical deployability. Finally, integrating cross-layer network vulnerabilities and post-quantum security considerations would strengthen the model's resilience against evolving cyber-physical threats.

## Author contributions

**Conceptualization:** Mian Imad Shah, Khaled Aldwoah.

**Formal analysis:** Mian Imad Shah.

**Investigation:** E. I. Hassan, Abdulghani Muhyi, W. Eltayeb Ahmed.

**Methodology:** Amjad Ali.

**Project administration:** Khaled Aldwoah.

**Resources:** W. Eltayeb Ahmed.

**Software:** Amjad Ali.

**Supervision:** E. I. Hassan.

**Validation:** Abdulghani Muhyi, Khaled Aldwoah.

**Writing – original draft:** Amjad Ali.

**Writing – review & editing:** Abdulghani Muhyi, Khaled Aldwoah.

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
