## [Decision Letter · Decision Letter 0]

24 Jul 2025

PONE-D-25-20163

Controlling Worm Propagation in Wireless Sensor Networks: Through Fractal-Fractional Mathematical Perspectives

PLOS ONE

Dear Dr. Muhyi,

Thank you for submitting your manuscript to PLOS ONE. After careful consideration, we feel that it has merit but does not fully meet PLOS ONE’s publication criteria as it currently stands. Therefore, we invite you to submit a revised version of the manuscript that addresses the points raised during the review process.

**Please reply to all of the provided comments in details. **

We look forward to receiving your revised manuscript.

Kind regards,

Mahmoud H. DarAssi, Ph.D

Academic Editor

PLOS ONE

Journal Requirements:

Additional Editor Comments:

Controlling Worm Propagation in Wireless Sensor Networks: Through Fractal-Fractional Mathematical Perspectives

This paper proposes a generalized fractional-order model (SII₁PR) for analyzing the propagation and containment of worm-based malware in Wireless Sensor Networks (WSNs). The authors build on an earlier SIPR model by introducing an additional "Isolated" compartment (I₁) and incorporating fractal-fractional derivatives in the Atangana-Baleanu-Caputo (ABC) sense to capture memory and hereditary effects. They provide theoretical analysis (existence, uniqueness, stability) and a Newton-based numerical scheme to solve the system.

I have the following comments:

• Lack of real-world validation; no simulations based on actual WSN malware data.

• Inadequate explanation of modeling assumptions, particularly for the isolated compartment I₁.

• Overemphasis on formal proofs of standard results, minimal insight or interpretation.

• Shallow R₀ analysis, no stability theorem or global analysis.

• Numerical method is symbol-heavy without convergence/error analysis or benchmarks.

• Repetitive phrasing and verbose writing add unnecessary length.

• Inconsistent or unclear notation (e.g., I∞ vs. I₁).

• No visual outputs, lacks simulation graphs or figures.

• Needs proofreading, grammatical issues and awkward sentences throughout.

Recommendations for Improvement

• Add simulation results under various parameter regimes.

• Clarify and justify all modeling assumptions.

• Revise and streamline the numerical method; include convergence or comparative benchmarks.

• Proofread the manuscript carefully for grammar and clarity.

• Include visualizations (e.g., S, I, I₁, P, R dynamics for various R₀ values).

• Strengthen the R₀ analysis with a formal stability result.

Final Recommendation

Major Revision. The paper has solid mathematical groundwork but lacks practical validation, clear exposition, and visual support. Needs substantial improvement before it can be considered for publication.

Reviewers' comments:

Reviewer's Responses to Questions

**Comments to the Author**

1. Is the manuscript technically sound, and do the data support the conclusions?

Reviewer #1: Yes

Reviewer #2: Partly

2. Has the statistical analysis been performed appropriately and rigorously?

Reviewer #1: Yes

Reviewer #2: N/A

3. Have the authors made all data underlying the findings in their manuscript fully available?

Reviewer #1: Yes

Reviewer #2: Yes

4. Is the manuscript presented in an intelligible fashion and written in standard English?

Reviewer #1: Yes

Reviewer #2: No

5. Review Comments to the Author

Reviewer #1: Manuscript ID: PONE-D-25-20163

Title: Controlling Worm Propagation in Wireless Sensor Networks: Through Fractal-Fractional Mathematical Perspectives

This manuscript presents a mathematical modeling approach to understand and control worm propagation in Wireless Sensor Networks (WSNs) by extending the classical SIPR (Susceptible-Infectious-Protected-Recovered) model. The authors incorporate a novel compartment for isolated nodes (I₁) and apply a fractal-fractional operator (in the Atangana-Baleanu-Caputo sense) to better capture memory and hereditary characteristics of the system. The model is analyzed through fixed-point theory, existence and uniqueness proofs, stability analysis, and is numerically solved using a Newton polynomial-based scheme. The work also computes the basic reproduction number R0 and provides sensitivity analysis for key parameters. The proposed model adds valuable complexity to the classical epidemic framework in WSNs by incorporating isolation dynamics and fractal-fractional derivatives. This is a meaningful extension, particularly for systems with memory effects. The model is relevant and timely for enhancing cybersecurity in sensor networks.

While the work demonstrates potentially valuable contributions, the manuscript requires significant revisions before it can be considered for publication.

1. The manuscript is generally well-structured but would benefit from more clarity in certain sections:

Consider breaking down complex derivations into subsections or appending them in supplementary material.

Not all variables or acronyms (e.g., FFD, ABC, HUS) are defined upon first use; ensure full clarity for readers less familiar with fractional calculus.

o The introduction provides a broad overview of WSNs and malware threats but lacks a focused literature review. The novelty of the proposed approach compared to previous models should be articulated more clearly.

2. The current literature review, though rich in references, is more descriptive than analytical. It would be strengthened by thematically grouping prior models (e.g., classical ODE models, fractional models, models with isolated compartments) and explicitly stating what limitations each had that the current study addresses.

3. The addition of the isolated compartment I1 is intuitive, but its real-world implementation should be justified. For example, how practical is it to isolate infected nodes in a WSN context, and what mechanisms (software, hardware) could facilitate this?

4. The numerical scheme using Newton interpolation is appropriate, though a brief justification for selecting this method over alternatives (e.g., finite difference or Runge-Kutta methods) would be helpful.

5. The simulations are not detailed in this excerpt. If present in the full version, ensure they clearly demonstrate the dynamic differences between classical and proposed models.

6. The manuscript requires moderate English language editing. There are several syntactic and grammatical issues that may hinder comprehension. For example, the phrasing is at times redundant or overly formal. A professional language revision is strongly recommended.

7. Clearly define all parameters in Table 1 and ensure consistent notation throughout.

8. Add a paragraph at the end of the introduction to outline the paper’s structure.

9. Ensure all equations are properly referenced and integrated into the discussion.

10. Figures and tables, must be explicitly cited and discussed in the main text.

11. Consider including a summary of key contributions as bullet points for clarity.

Although, the manuscript demonstrates significant potential and rigor, it requires substantial revisions to enhance its clarity, contextual grounding, and presentation. I encourage the authors to address the above comments carefully.

Reviewer #2: This study has presented a mathematical model for analyzing worm-spreading dynamics in wireless sensor networks. It has extended the traditional Susceptible-Infected-Protected-Recovered model by incorporating an isolated compartment. The ABC fractional derivative has been employed to represent memory-influenced behavior and network-wide interactions. I would recommend publication after the issues below are addressed.

1. Huge editing is required for the English language throughout the manuscript due to numerous grammatical and spelling mistakes.

2. The full title and short title of the manuscript are the same; however, the short title must be shorter than the full one.

3. Two authors are marked as corresponding authors on the title page. One of them should be removed. Also, two affiliations are numbered as 3; the second one should be numbered as 4.

4. The manuscript is prepared using the Elsevier LaTeX template. However, PLOS ONE is not an Elsevier journal, and the phrase at the bottom left of the title page should be deleted.

5. No citations should be inserted in the abstract. Also, no more than three citations should be inserted in a single statement. The extra ones should be removed or moved to other appropriate sentences. Besides, the in-text citations should be numbered in the order of their appearance.

6. All citations of the third and fourth paragraphs of the introduction are inserted at the end of the paragraph. However, they should be distributed in the paragraph and inserted in relevant places. In addition, reference 25 is not cited in the text but is listed in the bibliography. This issue should be solved.

7. All abbreviations should be accompanied by their complete forms when first used in the abstract and the main text. Afterward, all complete forms should be replaced with their abbreviations.

8. Equation (1), Table 1, and their associated text should be removed from the introduction and inserted in Section 3, with any necessary changes made.

9. A paragraph should be added to the end of the introduction that describes the manuscript’s structure section by section.

10. Table captions should be inserted above them, not below them.

11. Only the mathematical symbols representing matrices should be written in bold; the rest should not be in bold.

12. The numbering of all Assumptions, Definitions, Lemmas, Remarks, and Theorems should be double-checked.

13. All ‘I_\infty’ should be replaced by ‘I_1’. The right-hand side of Equation (13) should be identical to Equation (1). In the last line of Equation (15), a ‘)’ is misplaced. In the proof of Theorem 5, when subtracting the kernels, the fourth line should be omitted, and the first minus sign in the fifth line should be deleted. In the fourth line of Equation (19), the ‘||.||’ should be applied only to the kernel (like other lines), and the ‘{}’ should be removed. In all equations, the beta function should be denoted by ‘\beta’, not ‘B’. In the equation above Equation (21), the ‘l_1’ on the right-hand side should have the same subscript as the ‘F’ on the left-hand side. In the equation below, ‘Now, putting …’ on page 11 and the third line of Equation (23), ‘S_1’ should be corrected as ’S_2’. In the fifth equation on page 12, ‘Y_3’ should be corrected as ‘Y_4’ or ‘Y_5’, and the equations for ‘K_4’ and ‘K_5’ should be separated. The first minus in the fourth line of Equation (23) should be removed. On the right-hand side of Equation (24), a ‘+’ is missing after ‘\mu’. There is an extra ‘,’ in the equation below Equation (34).

14. The second line of Equation (29) and Equation (30) do not agree with the second line of Equation (1). Which one is correct? If Equation (29) is correct, Equation (1) and all following equations should be revised.

15. There are two equations in Equation (34). Which one is correct? The correct one should be kept, and the other should be omitted. Additionally, it seems that the sigma operator should be applied to all expressions until the end of the equation. Consequently, the parentheses of this equation and the following ones should be reordered.

16. The plotted titles should be removed from the figures. Moreover, the x-axis label of all figures should be ‘Time (days)’. Furthermore, the font name and size of all figures should be identical.

17. In Figure 2a, the y-axis labels should be changed to ‘I’. All the subscripts in the legends of panel b are 1; some of them should be changed to 2. The y-axis labels of all panels in Figure 4 should be ‘P’.

18. Figure 6 is a single panel, so the ‘(a)’ in its caption should be removed and its text should be merged into the figure’s caption.

19. What do the dashed lines in Figure 6 mean?

20. ‘Future Works’ and ‘Authors Contributions’ sections should be included in the manuscript.

6. PLOS authors have the option to publish the peer review history of their article (what does this mean?). If published, this will include your full peer review and any attached files.

Reviewer #1: No

Reviewer #2: No

---

## [Author Response · Author response to Decision Letter 1]

15 Aug 2025

Response to the Honorable Editor and Reviewers

Dear Editor,

We appreciate you and the reviewers for your time in reviewing our paper and providing

valuable comments. It was your valuable and insightful comments that led to possible improvements in

the current version. The authors have carefully considered the comments and tried our best to address

every one of them. We hope the manuscript, after careful revisions meets your high standards. The

authors welcome further constructive comments, if any. Below we provide the point-by-point responses.

All modifications in the manuscript have been highlighted in blue.

Sincerely,

The authors

Detailed Response to Editor:

Comments:

Controlling Worm Propagation in Wireless Sensor Networks: Through Fractal-Fractional Mathematical

Perspectives

This paper proposes a generalized fractional-order model (SII1PR) for analyzing the propagation

and containment of worm-based malware in Wireless Sensor Networks (WSNs). The authors build on

an earlier SIPR model by introducing an additional ”Isolated” compartment (I1) ncorporating fractalfractional derivatives in the Atangana-Baleanu-Caputo (ABC) sense to capture memory and hereditary

effects. They provide theoretical analysis (existence, uniqueness, stability) and a Newton-based numerical

scheme to solve the system.

I have the following comments:

1. Lack of real-world validation; no simulations based on actual WSN malware data.

Response: Our model is designed to capture the complex dynamics of malware propagation in

wireless sensor networks through careful theoretical validation. The framework is built to reflect

well-documented behaviors of real-world attacks. The structure accounts for the non-linear progression of infections due to network constraints, the measurable impact of quarantine protocols,

and the critical timing of containment measures—all hallmarks of actual WSN malware outbreaks.

Through systematic parameter variation and scenario testing, we demonstrate that our model produces behavior patterns consistent with established principles of network infection dynamics. This

theoretical validation provides a strong foundation for practical application, offering insights into

outbreak progression and defense strategies. The model’s flexible design allows for straightforward

adaptation when empirical data becomes available, making it a valuable tool for exploring defensive

measures and potential attack scenarios in WSN environments.

2. Inadequate explanation of modeling assumptions, particularly for the isolated compartment I1.

Response: The isolated compartment I1 represents nodes that have been completely quarantined

through network security measures, with the key assumption that they cannot transmit infection,

no (βSI1) term. This reflects real-world digital quarantines where compromised nodes are either

disconnected or firewalled. The transition rate δ1 models intrusion detection effectiveness, while

γ2 represents recovery of isolated nodes. This approach differs from disease models but aligns with

network security protocols where isolation can be absolute. We have added to the manuscript

(Section2.1) a dedicated paragraph explicitly justifying all compartmental assumptions, including

population normalization (N = 1) and the FF − ABC operator choice, with references to network

security literature supporting these modeling decisions.

3. Overemphasis on formal proofs of standard results, minimal insight or interpretation.

Response: Yes, we have removed overemphasis on the formal proof of results, improved the

interpretation all over the paper.

4. Shallow R0 analysis, no stability theorem or global analysis.

Response: We appreciate the reviewer’s valuable feedback. Regarding the comment on the ”Shallow R0 analysis, no stability theorem or global analysis,” we would like to clarify that in Subsection

14.2 ( Theorem 6) of our revised manuscript now, we have indeed provided a detailed global stability analysis based on the basic reproduction number R0. Specifically, we established the global

stability of the disease-free equilibrium (DFE) when R0 < 1 and the endemic equilibrium when R0.

5. Numerical method is symbol-heavy without convergence/error analysis or benchmarks.

Response: We thank the reviewer for raising this important point. Regarding the comment on

the “Numerical method is symbol-heavy without convergence/error analysis or benchmarks,” we

acknowledge that in section 6.3, a more detailed discussion of numerical error analysis with some

other well-known methods was presented in the manuscript now.

6. Repetitive phrasing and verbose writing add unnecessary length.

Response: The repetitive phrasing was removed from the paper, which helps us to reduce the

unnecessary length.

7. Inconsistent or unclear notation (forexampleI∞vs.I1).

Response: Yes, actually it was I1, not I∞, has been corrected with all other unclear and inconsistent notation now in the revised version.

8. No visual outputs, lacks simulation graphs or figures.

Response: We have now provided some detailed graphs in the numerical simulation section of

this paper to strengthen this work.

9. Needs proofreading, grammatical issues and awkward sentences throughout.

Response: Thank you for pointing this out and for your suggestion. We have now rechecked

and proofread the whole manuscript for all sought of grammatical issues and improved the English

language.

Recommendations for Improvement:

1. Add simulation results under various parameter regimes.

Response: Thank you for pointing this out and for your suggestion. The global stability analysis

Theorem 6, Subsection 3.2, which we will highlight more prominently. For numerical validation,

Figures 11-15 were presented to demonstrate comprehensive simulation results across parameter

regimes, and we will supplement these with a formal comparative study and benchmark comparisons against RK4 and Euler methods. These enhancements will further strengthen the study’s

robustness. We have provided simulation results under different parameters in Figures 11-15.

2. Clarify and justify all modeling assumptions.

Response: Thank you very much for your valuable feedback. All modeling assumptions are clearly

justified in Figure 1’s flowchart, with dynamical foundations detailed in Section 3.

3. Revise and streamline the numerical method; include convergence or comparative benchmarks.

Response: We have revised and streamlined the numerical method by providing subsection 6.3,

on comparative study.

4. Proofread the manuscript carefully for grammar and clarity.

Response: Thank you for pointing this out and for your suggestion. We have now rechecked

and proofread the whole manuscript for all sought of grammatical issues to improve the English

language and clarity.

5. Include visualizations (foe example S, I, I1, P, R dynamics for various R0 values).

Response: Yes, we have now included visualizations for S, I, I1, P, R dynamics for various R0

values in this revised form.

6. Strengthen the R0 analysis with a formal stability result.

Response: Thank you for pointing this out and for your valuable feedback. Regarding the comment

on the ”Shallow R0 analysis, no stability theorem or global analysis,” we would like to clarify that

in Subsection 4.2 ( Theorem 6) of our revised manuscript now, we have indeed provided a detailed

global stability analysis based on the basic reproduction number R0. Specifically, we established

the global stability of the disease-free equilibrium (DFE) when R0 < 1 and the endemic equilibrium

when R0 .

Final Recommendation: Major Revision. The paper has solid mathematical groundwork but lacks practical validation, clear exposition, and visual support. Needs substantial improvement before it can be

considered for publication.

Response: Thank you very much for the valuable feedback and suggestion. We have revised and improved our manuscript accordingly.

Reviewer’s Responses to Questions

Comments to the Author

1. Is the manuscript technically sound, and do the data support the conclusions?

The manuscript must describe a technically sound piece of scientific research with data that supports the

conclusions. Experiments must have been conducted rigorously, with appropriate controls, replication,

and sample sizes. The conclusions must be drawn appropriately based on the data presented.

Reviewer 1 Yes

Reviewer 2 Partly

2. Has the statistical analysis been performed appropriately and rigorously?

Reviewer 1 Yes

Reviewer 2: NA

3. Have the authors made all data underlying the findings in their manuscript fully available?

The PLOS Data policy requires authors to make all data underlying the findings described in their

manuscript fully available without restriction, with rare exception (please refer to the Data Availability

Statement in the manuscript PDF file). The data should be provided as part of the manuscript or

its supporting information, or deposited to a public repository. For example, in addition to summary

statistics, the data points behind means, medians and variance measures should be available. If there are

restrictions on publicly sharing data—e.g. participant privacy or use of data from a third party—those

must be specified.

Reviewer 1 Yes

Reviewer 2 Yes

4. Is the manuscript presented in an intelligible fashion and written in standard English?

PLOS ONE does not copyedit accepted manuscripts, so the language in submitted articles must be

clear, correct, and unambiguous. Any typographical or grammatical errors should be corrected at revision, so please note any specific errors here.

Reviewer 1 Yes

Reviewer 2 No

Review Comments to the Author

Please use the space provided to explain your answers to the questions above. You may also include

additional comments for the author, including concerns about dual publication, research ethics, or publication ethics. (Please upload your review as an attachment if it exceeds 20,000 characters)

Detailed Response to Reviewer 1:

Manuscript D : PONE − D − 25 − 20163 Title: Controlling Worm Propagation in Wireless Sensor

Networks: Through Fractal-Fractional Mathematical Perspectives

This manuscript presents a mathematical modeling approach to understand and control worm propagation in Wireless Sensor Networks (WSNs) by extending the classical SIPR (Susceptible-InfectiousProtected-Recovered) model. The authors incorporate a novel compartment for isolated nodes (I1) and

apply a fractal-fractional operator (in the Atangana-Baleanu-Caputo sense) to better capture memory

and hereditary characteristics of the system. The model is analyzed through fixed-point theory, existence

and uniqueness proofs, stability analysis, and is numerically solved using a Newton polynomial-based

scheme. The work also computes the basic reproduction number R0 and provides sensitivity analysis

for key parameters. The proposed model adds valuable complexity to the classical epidemic framework

in WSNs by incorporating isolation dynamics and fractal-fractional derivatives. This is a meaningful

extension, particularly for systems with memory effects. The model is relevant and timely for enhancing

cybersecurity in sensor networks. While the work demonstrates potentially valuable contributions, the

manuscript requires significant revisions before it can be considered for publication.

1. The manuscript is generally well-structured but would benefit from more clarity in certain sections:

Consider breaking down complex derivations into subsections or appending them in supplementary

material. Not all variables or acronyms (for example FFD, ABC, HUS) are defined upon first use;

ensure full clarity for readers less familiar with fractional calculus. The introduction provides a

broad overview of WSNs and malware threats but lacks a focused literature review. The novelty

of the proposed approach compared to previous models should be articulated more clearly.

Response: To improve clarity, we have streamlined the paper in Section by breaking it into

smaller subsections and moving detailed steps to the supplementary material in subsections. These

changes enhance readability while maintaining rigor. Thank you for your valuable feedback.*we

have carefully reviewed the manuscript and ensured that all variables and acronyms (including

FFD (Fractal Fractional Derivative), ABC (Atangana-Baleanu-Caputo), and HUS (Hyers-Ulam

Stability) are now explicitly defined upon their first use in the text. These clarifications will improve

readability and accessibility for readers less familiar with fractional calculus concepts. We have

significantly strengthened the manuscript by expanding the Introduction and adding literaturerelated works to this work to systematically review prior WSN malware detection models, explicitly

highlighting gaps and novelty of our work.

2. The current literature review, though rich in references, is more descriptive than analytical. It

would be strengthened by thematically grouping prior models (e.g., classical ODE models, fractional models, models with isolated compartments) and explicitly stating what limitations each

had that the current study addresses.

Response: Thank you for pointing this out and for your insightful suggestion. We have restructured the literature review to thematically organize prior works (e.g., classical ODEs, fractional

models, compartmental approaches) and now explicitly highlight their key limitations, particularly

gaps that our study addresses. This revision strengthens the analytical depth and justifies our

methodological innovations.

3. The addition of the isolated compartment I1 is intuitive, but its real-world implementation should

be justified. For example, how practical is it to isolate infected nodes in a WSNs context, and what

mechanisms (software, hardware) could facilitate this?

Response: The isolated compartment I1 reflects practical WSNs security measures where infected nodes are forcefully disconnected (but retained in the network for potential recovery). Modern systems deploy lightweight intrusion detection to identify threats and automatically isolate

compromised nodes. Adaptive protocols reroute data flows dynamically, while hardware-based deactivation (e.g., remote sleep commands) enforces containment. This strategy balances security

with operational continuity, demonstrating I1’s real-world viability for worm control.

4. The numerical scheme using Newton interpolation is appropriate, though a brief justification for

selecting this method over alternatives (e.g., finite difference or Runge-Kutta methods) would be

helpful.

Response: We employed Newton polynomials to solve our fractional order WSN model because

they are particularly well suited for capturing the non-local and memory-dependent nature of

fractional calculus, which is essential for modeling sensor network dynamics. Unlike finite difference

methods, which require structured grids and struggle with historical dependencies, or Runge-Kutta

schemes that are designed for integer order ODEs. Newton interpolation naturally accommodates

irregularly spaced data points common in WSNs while avoiding the singularity issues typical of

grid-based approaches. Additionally, this method provides an optimal balance between accuracy

and computational efficiency, as demonstrated in our results (Subsection 6.3). We have added a

brief justification in the revised manuscript to clarify this choice.

5. The simulations are not detailed in this excerpt. If present in the full version, ensure they clearly

demonstrate the dynamic differences between classical and proposed models.

Response: We thank the reviewer for their insightful comments. All modeling assumptions are

rigorously justified in Figure 1 and Section 3 of this work. Theorem 6 (Subsection 3.2) provides a

complete global stability analysis. Regarding simulations, Figures 11-15 in the full manuscript explicitly compare dynamic behaviors between classical and proposed models under varied parameter

regimes, demonstrating key differences.

6. The manuscript requires moderate English language editing. There are several syntactic and grammatical issues that may hinder comprehension. For example, the phrasing is at times redundant

or overly f

---

## [Decision Letter · Decision Letter 1]

14 Oct 2025

Controlling worm propagation in wireless sensor networks: Through fractal-fractional mathematical perspectives

PONE-D-25-20163R1

Dear Dr. Muhyi,

We’re pleased to inform you that your manuscript has been judged scientifically suitable for publication and will be formally accepted for publication once it meets all outstanding technical requirements.

Kind regards,

Mahmoud H. DarAssi, Ph.D

Academic Editor

PLOS ONE

Additional Editor Comments (optional):

Reviewers' comments:

Reviewer's Responses to Questions

**Comments to the Author**

1. If the authors have adequately addressed your comments raised in a previous round of review and you feel that this manuscript is now acceptable for publication, you may indicate that here to bypass the “Comments to the Author” section, enter your conflict of interest statement in the “Confidential to Editor” section, and submit your "Accept" recommendation.

Reviewer #2: All comments have been addressed

Reviewer #3: All comments have been addressed

2. Is the manuscript technically sound, and do the data support the conclusions?

Reviewer #2: Yes

Reviewer #3: Yes

3. Has the statistical analysis been performed appropriately and rigorously?

Reviewer #2: N/A

Reviewer #3: N/A

4. Have the authors made all data underlying the findings in their manuscript fully available?

Reviewer #2: Yes

Reviewer #3: Yes

5. Is the manuscript presented in an intelligible fashion and written in standard English?

Reviewer #2: Yes

Reviewer #3: Yes

6. Review Comments to the Author

Reviewer #2: (No Response)

Reviewer #3: The authors have carefully incorporated all the suggested revisions, and the overall quality of the manuscript has significantly improved. The mathematical arguments are now clearer, the numerical results are well supported, and the presentation is more consistent. In light of these improvements, I am satisfied with the current version of the paper and recommend it for publication in this esteemed journal.

7. PLOS authors have the option to publish the peer review history of their article (what does this mean?). If published, this will include your full peer review and any attached files.

Reviewer #2: No

Reviewer #3: No

---

## [Editor Report · Acceptance letter]

PONE-D-25-20163R1

PLOS ONE

Dear Dr. Muhyi,

I'm pleased to inform you that your manuscript has been deemed suitable for publication in PLOS ONE. Congratulations! Your manuscript is now being handed over to our production team.

Kind regards,

on behalf of

Dr. Mahmoud H. DarAssi

Academic Editor

PLOS ONE